# Homeostasis of branched-chain amino acids is critical for the activity of TOR signaling in *Arabidopsis*

**Pengfei Cao[1,2], Sang-Jin Kim[3], Anqi Xing[4], Craig A Schenck[4], Lu Liu[1], Nan Jiang[4], Jie Wang[2], Robert L Last[2,4], Federica Brandizzi[1,2,3]***

[1]MSU-DOE Plant Research Lab, Michigan State University, East Lansing, United States; [2]Department of Plant Biology, Michigan State University, East Lansing, United States; [3]Great Lakes Bioenergy Research Center, Michigan State University, East Lansing, United States; [4]Department of Biochemistry and Molecular Biology, Michigan State University, East Lansing, United States

**Abstract** The target of rapamycin (TOR) kinase is an evolutionarily conserved hub of nutrient sensing and metabolic signaling. In plants, a functional connection of TOR activation with glucose availability was demonstrated, while it is yet unclear whether branched-chain amino acids (BCAAs) are a primary input of TOR signaling as they are in yeast and mammalian cells. Here, we report on the characterization of an Arabidopsis mutant over-accumulating BCAAs. Through chemical interventions targeting TOR and by examining mutants of BCAA biosynthesis and TOR signaling, we found that BCAA over-accumulation leads to up-regulation of TOR activity, which causes reorganization of the actin cytoskeleton and actin-associated endomembranes. Finally, we show that activation of TOR is concomitant with alteration of cell expansion, proliferation and specialized metabolism, leading to pleiotropic effects on plant growth and development. These results demonstrate that BCAAs contribute to plant TOR activation and reveal previously uncharted downstream subcellular processes of TOR signaling.

**\*For correspondence:**
fb@msu.edu

**Competing interests:** The authors declare that no competing interests exist.

## Introduction

In eukaryotes, target of rapamycin (TOR) is a conserved master regulator of metabolic signaling that integrates nutrient, energy, hormone, growth and stress inputs with cell growth and metabolism (*Saxton and Sabatini, 2017*; *Kim and Guan, 2019*; *Shi et al., 2018*). In yeast and mammalian cells, TOR associates with different interactors to form two functional complexes, TOR complex 1 (TORC1) and TOR complex 2 (TORC2) (*Saxton and Sabatini, 2017*). The Arabidopsis genome encodes functional homologs of TOR (AtTOR) and two TOR-interactors: LST8 (AtLST8-1 and AtLST8-2) and a specific component of TORC1, RAPTOR (AtRAPTOR1A and AtRAPTOR1B) (*Shi et al., 2018*). The Arabidopsis homologs of TORC2-specific components have not been identified yet.

The yeast and mammalian TORC1 and TORC2 receive specific inputs and regulate distinct downstream processes. For example, sensors of leucine (Leu) and other amino acids provide input signals to TORC1 (*Saxton and Sabatini, 2017*; *Kim and Guan, 2019*), while mammalian TORC2 is primarily regulated by the insulin-PI3K signaling and is inhibited by mTORC1 effectors (*Saxton and Sabatini, 2017*; *Xie et al., 2018*). The current model of plant TOR signaling is built upon the three TORC1 components identified to date and envisions that TORC1 has essential functions, including promoting synthesis of proteins, nucleotides and lipids, and inhibiting autophagy for cell growth and proliferation (*Shi et al., 2018*).

In plants, TOR has assumed critical and specific roles, including involvement in phytohormone signaling pathways (i.e., auxin [*Li et al., 2017*; *Chen et al., 2018*; *Schepetilnikov et al., 2017*],

cytokinin [*Pfeiffer et al., 2016*], brassinosteroid [*Zhang et al., 2016*; *Xiong et al., 2017*] and abscisic acid [*Wang et al., 2018*]), as well as metabolic signaling (e.g., glucosinolates [*Malinovsky et al., 2017*], sulfur sensing [*Dong et al., 2017*]). Despite these advances, it is not well elucidated which inputs activate plant TOR signaling. In yeast and mammalian models, nutrient sensing of amino acids, especially the branched-chain amino acids (BCAAs) Leu, isoleucine (Ile) and valine (Val), is the primary input underlying activation of TOR signaling (*Saxton and Sabatini, 2017*; *Kim and Guan, 2019*). Studies in plants revealed a glucose-TOR signaling pathway that integrates light and sugar availability to control meristem activation in root and shoot (*Li et al., 2017*; *Chen et al., 2018*; *Pfeiffer et al., 2016*; *Xiong et al., 2013*), connecting TOR signaling to the generation and availability of photosynthates. A recent work identified isopropylmalate synthase 1 (IPMS1), a critical enzyme for Leu biosynthesis, as a suppressor of *lrx1*, a mutant that is defective in root hair development (*Schaufelberger et al., 2019*). Additionally, this research found that mutation of IPMS1 affects root development by reducing its sensitivity to chemical inhibition of TOR (*Schaufelberger et al., 2019*), suggesting alteration of Leu biosynthesis modifies the TOR network. Therefore, the level of conservation of TOR signaling input across kingdoms and in plant cells specifically is still a significant question.

Investigating the relationship between BCAAs and plant TOR signaling has been generally hindered by the innate capacity of plant cells to produce BCAAs (*Xing and Last, 2017*). In the chloroplast, the allosterically regulated enzymes threonine deaminase (TD), acetohydroxyacid synthase (AHAS) and IPMS are subjected to feedback inhibition, and contribute to BCAA homeostasis (*Xing and Last, 2017*). Mammalian cells are unable to synthesize BCAAs; consequently, starvation and repletion of BCAAs in the culture serum can be effective in stimulating TOR signaling (*Sarbassov et al., 2004*; *Jacinto et al., 2004*). In contrast, for plant cells, exogenous feeding of a limited concentration of amino acids to the growth medium may not perturb BCAA homeostasis and trigger detectable activation of TOR signaling (*Xiong et al., 2013*). Moreover, supplementation of a single or a combination of multiple amino acids may confer intertwined feedback inhibition to the biosynthetic pathway, and lead to unpredictable disruption of amino acid homeostasis (*Galili et al., 2016*; *Maliga, 1984*). Additionally, because TOR is the main negative regulator of autophagy and promotes protein synthesis, either chemical inhibition of BCAA biosynthesis and TOR activity or mutation of TOR signaling components does not specifically affect BCAAs but leads to substantial increases in almost all types of amino acids in plant cells (*Ren et al., 2012*; *Zhao et al., 2018*; *Mubeen et al., 2018*; *Moreau et al., 2012*; *Caldana et al., 2013*). In light of these considerations, it is therefore not surprising that little is known about the functional and physiological consequences of an up-regulation of TOR signaling in plants, especially at a subcellular level. In yeast and mammalian cells, induced activation of TOR governs numerous cellular processes, most evidently the inhibition of autophagy by TORC1 and the reorganization of actin cytoskeleton by TORC2 (*Saxton and Sabatini, 2017*). Although the mechanistic basis is not fully understood, it has been proposed that mTORC2 regulates reorganization of actin cytoskeleton through the AGC family protein kinases and Rho signaling (*Xie et al., 2018*). Plants express AGC family-like homologs with specialized functions (*Garcia et al., 2012*) and Rho of Plants (ROP) members that signal to the cytoskeleton system (*Garcia et al., 2012*; *Feiguelman et al., 2018*). Moreover, recent studies reported an interaction between ROP2 and TOR in the context of TOR-auxin crosstalk (*Li et al., 2017*; *Schepetilnikov et al., 2017*). Nonetheless, direct functional connections between TOR signaling, cytoskeleton in plant growth and development are yet to be established (*Ren et al., 2012*; *Moreau et al., 2012*; *Deprost et al., 2007*; *Salem et al., 2018*; *Salem et al., 2017*; *Anderson et al., 2005*; *Deprost et al., 2005*; *Xiong and Sheen, 2012*).

The vacuole and the endoplasmic reticulum (ER) are the two organelles of largest membrane extension in plant cells (*Zhang et al., 2014*; *Stefano et al., 2014*). The central vacuole fulfills essential cellular functions such as providing turgor pressure, protein turnover and metabolite storage (*Zhang et al., 2014*). The ER is the gateway to the secretory pathway and a membrane network that weaves through nearly all the other types of organelles (*Stefano et al., 2014*; *Wu et al., 2018*). Despite of the essential roles of the vacuole and the ER (*Zhang et al., 2014*; *Stefano et al., 2014*), mechanisms underpinning their morphogenesis are still largely undefined although homotypic membrane fusion is known to be required (*Zhang et al., 2014*; *Cui et al., 2019*; *Stefano et al., 2012*). It is also yet unclear whether and how the dynamics of plant ER and vacuoles are controlled in response to different developmental and environmental clues. The dynamics of endomembrane

system in plant cells are mechanistically different from mammalian cells. In the latter, the ER morphology, organization and dynamics are mainly driven by microtubules and microtubule-related motor proteins (*Westrate et al., 2015*). In net contrast, the plant ER is primarily anchored to and mobilized by the actomyosin system (*Zhang et al., 2014*; *Stefano and Brandizzi, 2018*; *Ueda et al., 2010*). Plant vacuoles are also in close proximity to the actin cytoskeleton (*Zhang et al., 2014*). Furthermore, interactions between plant ER and vacuolar membranes with bundled actin filaments form ER strands and *trans*-vacuolar strands (TVSs) (*Ueda et al., 2010*). The regulatory mechanisms of these membrane structures are also unknown.

In this work, we report on the characterization of *eva1* (*ER, vacuole and actin 1*), a mutant with defects in vacuole morphogenesis and organization of actin filaments and endomembranes, which are associated mainly with actin in plant cells (*Ueda et al., 2010*). The *eva1* mutation is a loss-of-function allele of *IPMS1*, which encodes the first committed enzyme of Leu biosynthesis, resulting in elevated free Val levels. Through phenotypic and functional analyses of *eva1* and a series of other mutants of BCAA biosynthesis and TOR signaling, we demonstrate that the subcellular phenotypes of *eva1* specifically hinge upon up-regulation of TOR signaling, which in turn affects organization of actin and endomembranes, and plant development. Therefore, by focusing on mutants with constitutive TOR signaling mis-regulation due to altered endogenous BCAA levels, we demonstrated that plant TOR signaling is linked to BCAAs and is critical for the homeostasis of actin, endomembranes and growth. The broader implications of these findings are that, despite the acquisition of specialized functions of TOR signaling in plants, the activating inputs of TOR signaling and the subcellular consequences of TOR signaling mis-regulation are conserved across eukaryotes.

## Results

### Identification of a mutant with defects in vacuole morphogenesis

We pursued a confocal microscopy-based screen on an EMS-mutagenized population to identify mutants with defects in the subcellular distribution of a GFP-tagged tonoplast intrinsic protein (TIP), GFP-δTIP (*Avila et al., 2003*; *Cutler et al., 2000*). We focused on *eva1*, a mutant characterized by severe defects in vacuole morphology early in development. During the first 10 days after germination, in wild-type (WT) cotyledon epidermal cells, small vacuoles undergo membrane fusion to form a single large central vacuole (*Zhang et al., 2014*) (*Figure 1a*). In contrast, 10 day old *eva1* cotyledon epidermal cells displayed numerous additional vacuolar structures that vary in shape and size (*Figure 1b*). To further characterize *eva1* vacuolar phenotypes, we focused on two prominent vacuolar structures that are rarely observed in wild type: TVSs and presumably unfused vacuoles. As we introduced, TVSs are strands formed upon association between vacuolar membrane and bundled actin filaments (*Ueda et al., 2010*), which were greatly enhanced in number, length and thickness in *eva1* (*Figure 1—figure supplement 1a–d*). Besides, we defined presumably unfused vacuoles as spherical structures that are isolated from the large central vacuole and have diameter >5 µm. Both two vacuolar phenotypes were attenuated in 20 day old *eva1* cotyledons, which closely resembled WT (*Figure 1c,d*). The *eva1* vacuole phenotypes were verified in 10 day old *eva1* cotyledons expressing γTIP-YFP (*Nelson et al., 2007*), which labels the large central vacuole and other vacuolar structures not marked by GFP-δTIP (*Gattolin et al., 2010*) (*Figure 1—figure supplement 1e–h*). These results support that the tonoplast organization and vacuolar morphology are compromised in *eva1* in early stages of growth independently from the tonoplast marker used for the analyses.

We next aimed to identify the causative mutation in *eva1*. Bulked segregant analysis and whole-genome resequencing narrowed down the *eva1* mutation to a G-to-A transition in *IPMS1* (*AT1G18500*) causing an aspartate (Asp)-to-asparagine (Asn) residue substitution (*Figure 1e,f*; *Figure 1—figure supplement 2a*). IPMS1 catalyzes condensation of 2-oxoisovalerate and acetyl-CoA into 2-isopropylmalate, the committed step for Leu biosynthesis (*de Kraker et al., 2007*; *Field et al., 2004*) (*Figure 2a*). Homology modeling of IPMS1 predicted that the mutated Asp228 is located in the acetyl-CoA binding surface near the pocket for 2-oxoisovalerate substrate (*Figure 1—figure supplement 2b,c*). In addition to *eva1*, we used three other *IPMS1* alleles that had been characterized: two recessive loss-of-function mutants, *ipms1-4* and *ipms1-5*, and a gain-of-function *ipms1-1D*, with a point mutation that impairs allosteric regulation (*Xing and Last, 2017*; *de Kraker et al., 2007*) (*Figure 1e*). 10 day old *eva1*, *ipms1-4* and *ipms1-5* seedlings exhibited similar delay in

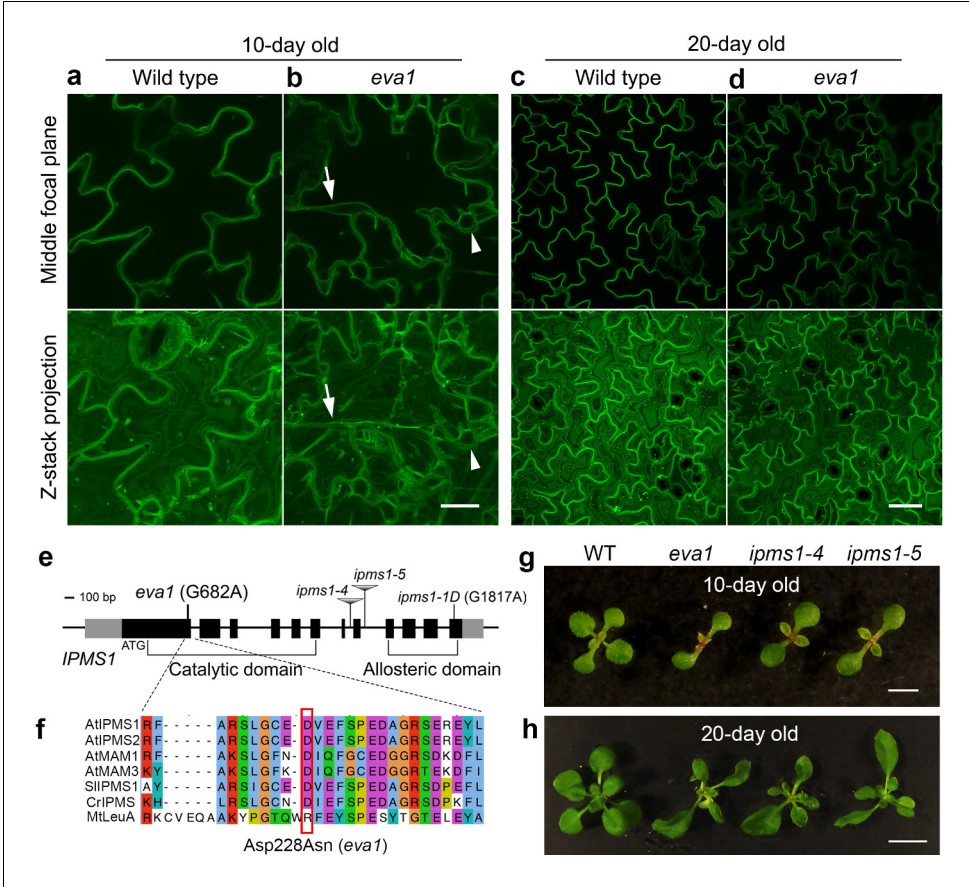

**Figure 1.** Identification of a mutant with defects in vacuole morphogenesis. (a – d) Confocal images of cotyledon epidermal cells expressing tonoplast marker GFP-δTIP in 10 day old (**a and b**) and 20 day old (**c and d**) wild type and *eva1*. The top panels present single images of the middle focal plane of the epidermal cells. The bottom panels present Z-stack maximal projections, which is a stack of about 20 single images with 20 μm intervals that fully span the top-to-bottom Z-axis of the epidermal cells. Arrows point to TVSs and arrowheads indicate presumably small vacuoles, which are prominent in *eva1*. Scale bar in (**a and b**), 20 μm. Scale bar in (**c and d**), 50 μm. (**e**) Genomic structure of *IPMS1* (AT1G18500). Gray boxes: UTRs; black boxes: exons; lines: introns. (**f**) Amino acid sequence alignment of IPMS1 homologs using T-COFFEE in Jalview. Amino acids are grouped by color with ClustalX based on their similarity of physicochemical properties. At, *Arabidopsis thaliana*; Sl, *Solanum lycopersicum*; Cr, *Chlamydomonas reinhardtii*; Mt, *Mycobacterium tuberculosis*. The amino acid substitution of *eva1* is denoted by a red box. (**g and h**), Photographs of 10-day old (**g**) and 20-day old (**h**) plants. Scale bar in g, 0.5 cm. Scale bar in h, 1 cm.

The online version of this article includes the following figure supplement(s) for figure 1:

**Figure supplement 1.** *eva1* mutants show vacuolar phenotypes of TVSs and presumably unfused vacuoles.
**Figure supplement 2.** The causal mutation of *eva1* mutant is mapped to *IPMS1*.

emergence of true leaves (*Figure 1g*). These growth and developmental phenotypes, as well as the subcellular phenotypes, were attenuated by 20 days of growth (*Figure 1h*). The presence of the *eva1* phenotypes in the *eva1* × *ipms1-5* F1 progeny confirmed allelism of *eva1* to *IPMS1* (*Figure 1— figure supplement 2d–f*). Together, these results support that the *eva1* vacuole and plant growth phenotypes are correlated to a loss of functional IPMS1, which has a consistent subcellular impact on early stages of growth.

## *eva1* plants over-accumulate val

The role of IPMS1 in BCAA biosynthesis has been well characterized as directing flux towards Leu biosynthesis, and away from the competing product Val (*Xing and Last, 2017*; *de Kraker et al., 2007*; *Field et al., 2004*) (*Figure 2a*). The Arabidopsis genome encodes two IPMS isoforms: IPMS1

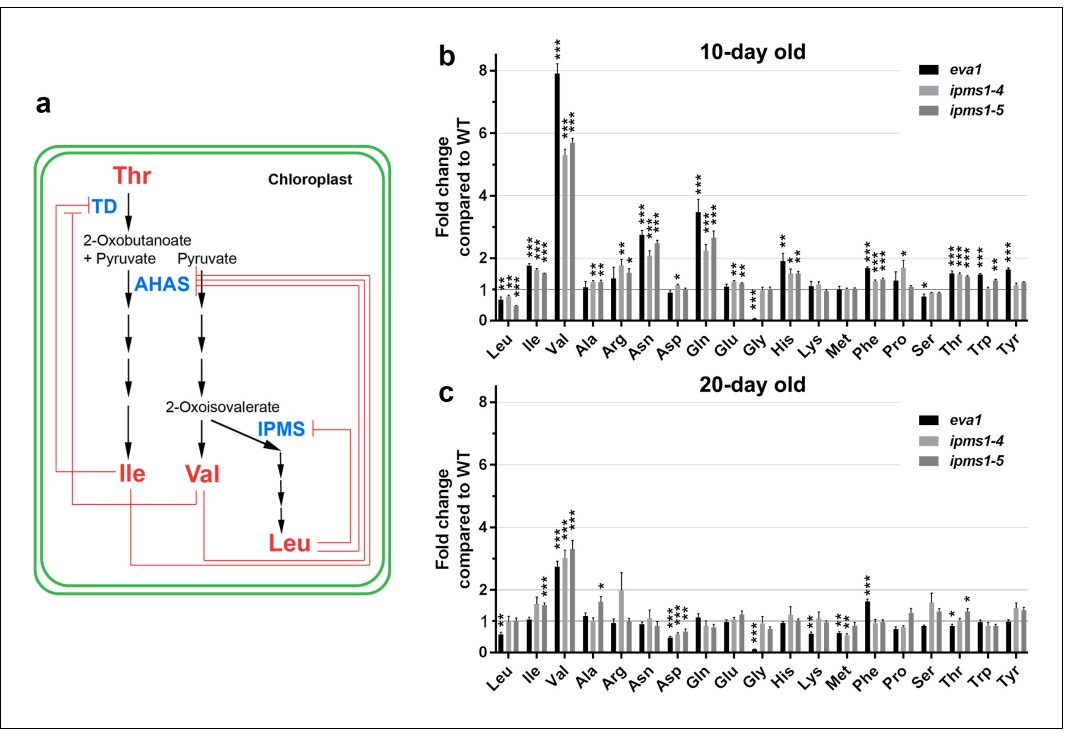

**Figure 2.** Amino acid profiling of *IPMS1* loss-of-function mutants. (a) Schematic of the BCAA biosynthetic pathway in the chloroplast. Red lines show known feedback inhibitions of enzymes by end products. (b and c) Fold changes of each free amino acid in *eva1*, *ipms1-4* and *ipms1-5* compared to wild type. Values are mean ± SEM. For 10 day-old sampling, n = 7 for WT, n = 5 for *eva1*, n = 8 for *ipms1-4*, n = 7 for *ipms1-5*. For 20 day-old sampling, n = 6 for each genotype. The asterisks indicate significant differences compared to the wild type (*p≤0.05, **p≤0.01, ***p≤0.001, unpaired *t* test).

The online version of this article includes the following source data for figure 2:

**Source data 1.** The BCAA biosynthesis pathway is up-regulated in young seedlings of *eva1* and *IPMS1* loss-of-function mutants.

mRNA accumulates to higher levels than IPMS2 mRNA through most stages of plant growth (*de Kraker et al., 2007*). An earlier work examined two-week old plants and found that Val and Ile were increased in both *ipms1-4* and *ipms1-5* but Leu was decreased in *ipms1-4* and increased in *ipms1-5* (*Xing and Last, 2017*). To determine the impact of the *eva1* mutation on amino acid homeostasis at earlier stages of growth, we conducted free amino acid analysis of 10- and 20 day old WT and *ipms1* mutants. Notably, we found that *eva1*, *ipms1-4* and *ipms1-5* had similar increases in Val and total BCAAs and decreases in Leu, consistent with our findings that *eva1* is a loss-of-function allele of *IPMS1* (*Figure 2b,c*; *Figure 2—source data 1*). In addition, in these mutants we found similar changes of Asp-derived amino acids (threonine-Thr, methionine-Met, lysine-Lys and Ile) and aromatic amino acids (phenylalanine-Phe, tryptophan-Trp and tyrosine-Tyr) (*Figure 2b,c*). Consistent with a disappearance of the subcellular phenotypes of the mutants during growth (*Figure 1a - d*), the impact of *ipms1* mutations on amino acid homeostasis was mitigated at 20 days of growth, with the fold change of Val becoming smaller in the mutants *versus* WT, and the types of amino acids significantly changed in the mutants compared to WT becoming fewer (*Figure 2b,c*). Taken together, these data indicate that *eva1* is a loss-of-function mutant of *IPMS1* equivalent to *ipms1-4* and *ipms1-5* and that the alteration in BCAA levels is most notable for increased in Val levels.

## Disruption of BCAA homeostasis leads to pleiotropic defects in plant growth and development

Next, we asked whether the transient changes in BCAA accumulation and vacuole morphology affected early plant growth. At 10 days following germination, the *IPMS1* loss-of-function mutants displayed retardation of growth and development (*Figure 1g,h*), showing approximately 30–40%

decrease in aerial tissue fresh weight and 40–50% decrease in primary root length compared to WT (*Figure 3—figure supplement 1a,b*). Propidium iodide staining showed a strikingly delayed formation of root hairs in *ipms1* mutants compared to WT (*Figure 3—figure supplement 2a*), which was accompanied by increases in both cell length and number in the elongation zone (*Figure 3—figure supplement 2b,c*). Meanwhile, in *ipms1* meristem has increased cell number but reduced cell length (*Figure 3—figure supplement 2c,d*). At 20 days of growth, the difference in fresh weight between *ipms1* alleles and WT became not significant, though the primary roots of the *ipms1* mutants were still slightly shorter than WT (*Figure 3—figure supplement 1c,d*). In contrast, two independent lines of dominant *ipms1-1D* feedback-insensitive mutant, which have small Val decrease and Leu increase (*Xing and Last, 2017*), exhibited indistinguishable primary root elongation, but increased fresh weight compared to WT (*Figure 3—figure supplement 1a–d*). Additionally, we did not observe notable difference between six-week old WT and *IPMS1* loss-of-function mutants growing in soil (*Figure 3—figure supplement 1e*). The transient retardation of overall plant growth of *IPMS1* loss-of-function mutants correlated with the emergence-and-disappearance period of both vacuole morphology and BCAA homeostasis perturbation phenotypes (*Figure 1a - d*; *Figure 2b,c*).

We then examined the development of cotyledons, which constituted most of the aerial tissue for amino acid profiling and were used for confocal microscopy analyses. Cotyledons of *ipms1* mutants were thicker and larger than WT (*Figure 3a - c*). Despite a delay of true leaf emergence (*Figure 1g*), the expanded first pair of true leaves in these mutants were larger than WT (*Figure 1h*). Analyses of chloroplast ultrastructure revealed an absence of connecting stroma thylakoids and a reduction of thylakoid length *ipms1* alleles compared to WT (*Figure 3d,e*). Additionally, we noticed purple pigmentation in 10 day old *IPMS1* loss-of-function mutants, particularly in cotyledon petioles and emerging true leaves (*Figure 3—figure supplement 3a*). Anthocyanin extraction and measurement confirmed that these mutants contained higher levels of total anthocyanins compared to WT (*Figure 3—figure supplement 3b,c*). These results indicate that the growth of certain tissues of the *ipms1* mutants is particularly promoted but the overall plant growth and development are temporarily inhibited.

## The organization of ER network and actin cytoskeleton is altered in *eva1*

To gain more insights into the *eva1* vacuolar phenotypes, we extended our analyses to other endomembrane compartments. The endoplasmic reticulum (ER) is the most extensively distributed organelle of the plant secretory pathway, and it is closely associated with several other membrane-bound organelles, including the vacuole (*Ueda et al., 2010*). In the *eva1* mutant, the ER luminal marker ERYK (*Nelson et al., 2007*) revealed a pronounced appearance of the cortical ER network with strikingly thickened strands compared to WT (*Figure 4a*; see arrows). The thickened ER strands did not completely overlap with the TVSs (*Figure 4—figure supplement 1*). High-magnification confocal microscopy images of the cortical ER revealed a pronounced cisternation in *eva1* compared to WT (*Figure 4b*). Quantitative analyses of the surface area occupancy of the ER in the total field of view confirmed these observations (i.e., larger ER-occupied area in *eva1* compared to WT) (*Figure 4d*). The appearance of the Golgi apparatus, which in plant cells is organized in disperse stacks of cisternae in close association with the ER (*Brandizzi and Barlowe, 2013*), also was abnormal. Indeed, the Golgi marker GFP-CASP (*Renna et al., 2005*) revealed increased clustering and higher abundance of Golgi stacks at the cell cortex in *eva1* compared to WT (*Figure 4—figure supplement 2*). Next, we examined secretion to the apoplast with the bulk flow marker SEC-RFP (*Faso et al., 2009*). We found no intracellular retention of the marker in *eva1* (*Figure 4—figure supplement 3*), as it would be expected for mutants with defects in secretion, an important function of the endomembrane system (*Renna et al., 2013*; *Renna et al., 2018*). These results and the absence of retention of the vacuolar marker in the ER (*Figure 4—figure supplement 1*) document that the morphology of the vacuole, organization of the Golgi and the ER network are markedly affected by the *eva1* mutation, while bulk-flow secretion is unaffected.

Collectively, the root-related defects of the *ipms1* mutants, including delayed formation of root hairs and reduced number of lateral roots (*Figure 3—figure supplement 2*; *Figure 4—figure supplement 4*), are reminiscent of mutants with impaired actin depolymerization or promoted actin bundling (*Ketelaar et al., 2004*; *Deeks et al., 2005*), consistent with the possibility that reorganization of actin cytoskeleton may be causative of the observed developmental phenotypes. Furthermore,

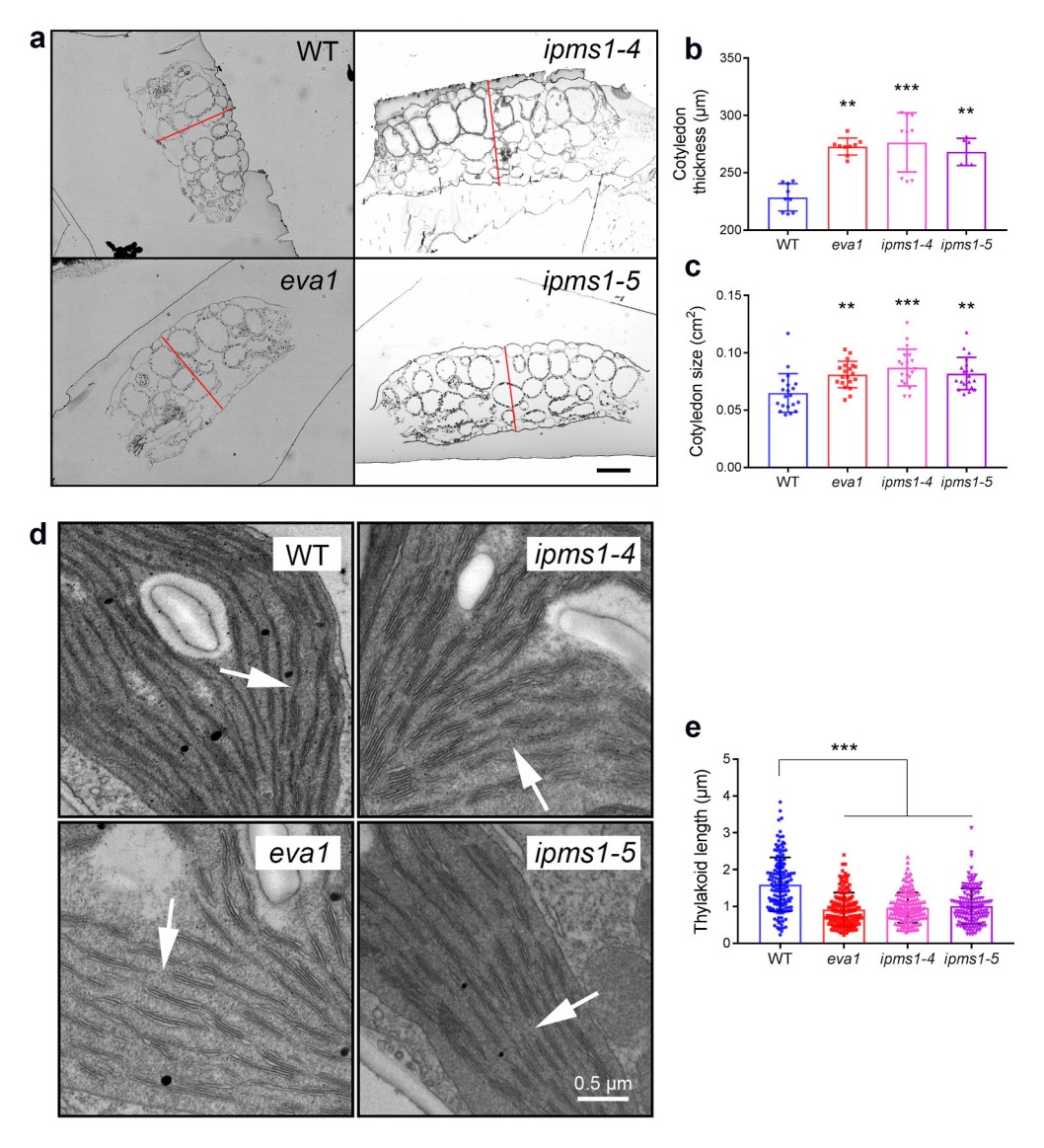

**Figure 3.** Mutants of *IPMS1* show defects in cotyledon architecture and chloroplast ultrastructure. (**a**) Light microscopic images of cotyledon cross sections. Cotyledon thickness is denoted by red lines. Scale bar, 100 μm. (**b**) Measurement of cotyledon thickness. n = 9 for WT, *eva1* and *ipms1-4*; n = 6 for *ipms1-5*. (**c**) Measurement of cotyledon size. n = 20 for each genotype. (**d**) Representative transmission electron microscopy images of chloroplasts. Arrows point to connecting stroma thylakoids that are existing in WT and absent in mutants. Scale bar, 0.5 μm. (**e**) Measurement of thylakoid length. Five cotyledons from each genotype were sampled for imaging, and at least 30 stroma thylakoids were measured in each sample (n ≥ 150). For all graphical representations of data, columns are mean ± SD. The asterisks indicate significant differences of each mutant compared to wild type (***p≤0.001, **p≤0.01, unpaired *t* test).

The online version of this article includes the following figure supplement(s) for figure 3:

**Figure supplement 1.** Mutations of *IPMS1* affect plant growth at early stage.
**Figure supplement 2.** Root tip staining and analyses.
**Figure supplement 3.** Accumulation of anthocyanins in *IPMS1* loss-of-function mutants.

because the establishment and maintenance of the TVSs, ER network and Golgi subcellular distribution are dependent on the actin cytoskeleton (*Ueda et al., 2010*), we hypothesized that the organization of actin cytoskeleton may be altered in *eva1*. Indeed, confocal microscopy in cells expressing the actin filament (F-actin) marker YFP-ABD2 (*Sheahan et al., 2004*) revealed coalescence of actin

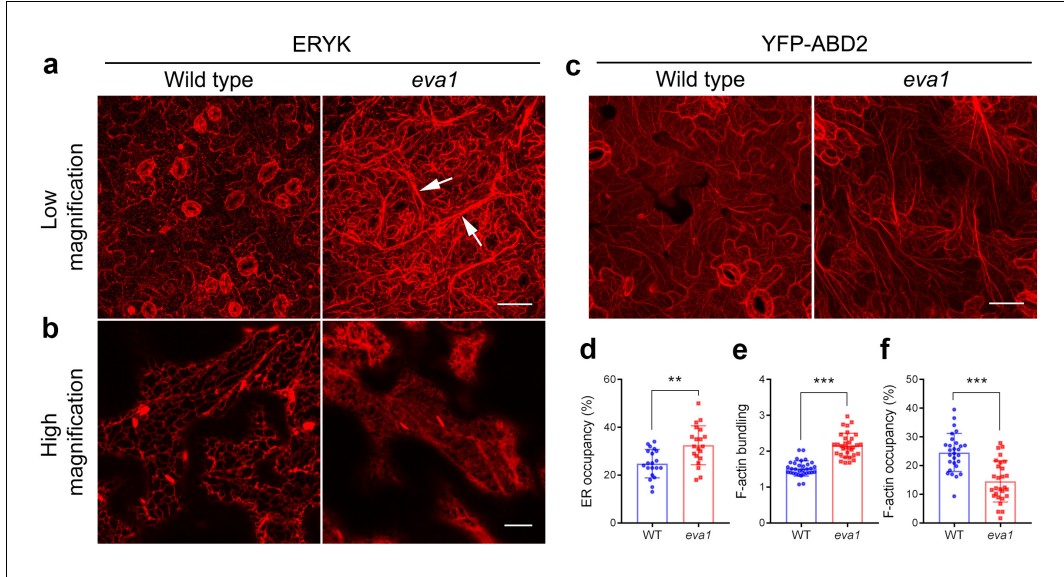

**Figure 4.** Mutation of *IPMS1* affects the ER morphology and the F-actin organization. (**a** and **b**) Confocal images of the wild type and *eva1* cotyledon epidermal cells expressing ER marker ERYK. (**a**) Low-magnification Z-stack projection images show the ER morphology in *eva1* is altered, featuring longer and more thickened ER strands, as the arrows indicate. Scale bar, 50 µm. (**b**) High-magnification single-plane images show enlarged ER sheets in *eva1* mutant. Scale bar, 10 µm. (**c**) Confocal images of the wild type and *eva1* cotyledon epidermal cells expressing F-actin marker YFP-ABD2. Scale bar, 50 µm. **d**), Quantification of ER occupancy, which measures the percentage of the area occupied by ER in the total field of view (n = 20 for each genotype). Single-plane images were used for the quantification. Columns show mean ± SD. The asterisks indicate significant differences (**p≤0.01, unpaired *t* test). (**e** and **f**) Quantitative evaluation of the F-actin organization using two parameters. Quantification of skewness (**e**) indicates higher level of F-actin bundling in *eva1* compared to wild type (n = 32 for each genotype); measurement of density (**f**) percentage suggests lower occupancy of F-actin in *eva1* compared to wild type (n = 28 for each genotype). Z-stack projection images were used for the quantification. Columns show mean ± SD. The asterisks indicate significant differences (**p≤0.01 and ***p≤0.001, unpaired *t* test).

The online version of this article includes the following figure supplement(s) for figure 4:

**Figure supplement 1.** The ER strands and *trans*-vacuolar strands are partially overlapping structures.

**Figure supplement 2.** The number and distribution of Golgi are altered in *eva1* mutant.

**Figure supplement 3.** The *eva1* mutation does not affect secretion of a bulk flow marker to the apoplast.

**Figure supplement 4.** Compared to wild type, *IPMS1* loss-of-function mutants are less sensitive to Lat B.

cables compared to WT (*Figure 4c*). Furthermore, quantitative analyses of actin organization identified higher skewness, suggesting enhanced bundling, and lower density, suggesting decreased occupancy of F-actin in the cytoplasm in *eva1* compared to WT (*Figure 4e,f*). These results imply that the prominent phenotypes of the endomembranes in *eva1* may be due to their connections with F-actin, whose organization is largely altered in the *eva1* mutant.

We next sought to validate this hypothesis by testing the sensitivity of the *ipms1* alleles to the F-actin depolymerizing reagent latrunculin B (Lat B) (*Cao et al., 2016*). The primary root length of 10 day old *ipms1-4* and *ipms1-5* was approximately 50% of WT (*Figure 4—figure supplement 4a,c*). Seedlings of all genotypes were then transferred to medium containing DMSO or 50 nM or 100 nM Lat B in DMSO. After another 8 days, we found that the Lat B treatment promoted the formation of lateral roots in WT seedlings, but not in *ipms1* alleles (*Figure 4—figure supplement 4b*). Additionally, the primary root length of *ipms1-4* and *ipms1-5* was approximately 65% compared to WT on DMSO medium; however, this difference was reduced in the presence of increasing levels of Lat B in the growth medium (i.e., 80% to WT on 50 nM Lat B, and not significantly different from WT on 100 nM Lat B) (*Figure 4—figure supplement 4d*). These results demonstrate that the *ipms1* alleles are less sensitive to F-actin depolymerization compared to WT, supporting a functional connection between the disruption of *IPMS1* and altered organization of the actin cytoskeleton.

## The *eva1* vacuolar phenotypes are rescued by PI3K/TOR dual inhibitors and partially recovered by disruption of F-actin

To gain insights into the mechanisms by which *eva1* defects in BCAA biosynthesis led to alteration of the organization of subcellular structures, we employed chemicals known to alter the vacuolar morphogenesis and cytoskeleton integrity. We hypothesized that the persistence of small vacuoles in *eva1* could be due to delayed vacuole membrane fusion during vacuole morphogenesis. To test this, we first employed wortmannin (Wm), an inhibitor of phosphoinositide 3-kinases (PI3Ks) that disrupts the balance of phosphoinositides and promotes homotypic tonoplast fusion (*Zheng et al., 2014*; *Wang et al., 2009*; *Mayer et al., 2000*). We found that treatment of 10 day old WT and *eva1* seedlings for two hours suppressed the *eva1* phenotypes (*Figure 5a - d*; *Figure 5—figure supplement 1*). The effects of Wm were mirrored by treatment with another PI3K inhibitor, LY294002 (*Zheng et al., 2014*) (*Figure 5—figure supplement 1e–h*). We then investigated a relationship between TVSs and integrity of the cytoskeleton in *eva1*. After a two-hour treatment with Lat B, we found that TVSs disappeared but the small vacuoles persisted in *eva1* cotyledon epidermal cells (*Figure 5e,f*). By contrast, a two-hour treatment with oryzalin, a microtubule disrupting reagent (*Zheng et al., 2014*), did not lead to discernable change of vacuole morphology (*Figure 5g,h*). Together these results indicate that the presumably unfused vacuole and enhanced TVS phenotypes in *eva1* are both responsive to Wm and LY294002, but only the enhanced TVS phenotype is related to the verified reorganization of F-actin.

## Loss of function of IPMS1 leads to up-regulation of TOR activity

Through chemical interventions, we confirmed that homotypic membrane fusion and F-actin bundling are two processes directly involved in the *eva1* Leu biosynthetic mutant phenotypes (*Figure 5*). This creates a quandary given that the role of IPMS1 in chloroplast BCAA biosynthesis is both functionally disconnected with – and spatially isolated from – the endomembrane compartments and

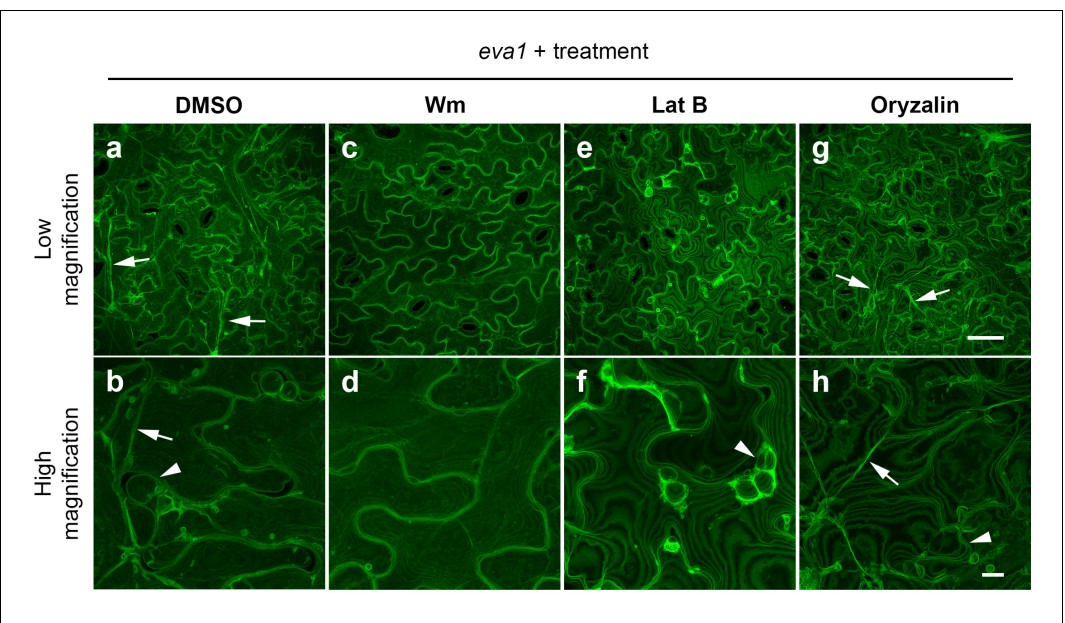

**Figure 5.** Chemical interventions can fully or partially rescue the vacuolar mutant phenotypes of *eva1*. (a – h) Confocal images of cotyledon epidermal cells expressing GFP-δTIP from 10 day old *eva1* plants. Images were acquired after 2 hr treatment of DMSO (a and b), wortmannin (Wm; c and d), latrunculin B (Lat B; e and f), or oryzalin (g and h). The top panel (a), (c), (e, and g) presents images of lower magnification with a scale bar of 50 μm. The bottom panel (b), (d), (f, and h) presents images of higher magnification with a scale bar of 10 μm. Arrowheads suggest presumably unfused vacuolar structures and arrows pinpoint enhanced TVSs. All the images are Z-stack maximal projections.

The online version of this article includes the following figure supplement(s) for figure 5:

**Figure supplement 1.** Chemical treatment with PI3K/TOR dual inhibitors wortmannin and LY294002.

actin cytoskeleton. Although the functions of Wm and LY294002 in inhibiting PI3Ks and promoting homotypic vacuolar membrane fusion have been established in plant cells (*Cui et al., 2019*; *Zheng et al., 2014*; *Wang et al., 2009*; *Marshall and Vierstra, 2018*), in mammalian cell studies these chemicals have been used to inhibit TOR signaling (*Sarbassov et al., 2004*; *Brunn et al., 1996*). This is because TOR belongs to the phosphoinositide kinase-related kinase (PIKK) family, whose members share similar kinase domains with PI3Ks (*Andrs et al., 2015*). Indeed, Wm and LY294002 are effective inhibitors of mammalian TOR (*Brunn et al., 1996*), and thus are considered as PI3K/TOR dual inhibitors (*Benjamin et al., 2011*). These considerations and our results led us to hypothesize that the effects of Wm and LY294002 in suppressing the *eva1* vacuole phenotypes could be related to TOR inhibition.

To test this hypothesis, we employed two TOR inhibitors with high selectivity for TOR over PI3Ks: AZD-8055 and Torin2 (*Benjamin et al., 2011*; *Liu et al., 2011*; *Chresta et al., 2010*), which also effectively inhibit plant TOR (*Li et al., 2017*; *Wang et al., 2018*; *Dong et al., 2017*; *Pu et al., 2017*). We transferred 10 day old WT and *eva1* seedlings to liquid growth medium containing 5 μM AZD-8055. Compared to untreated samples, WT cells did not exhibit significant changes in the morphology of the central vacuole and the few thin TVSs after 2 or 4 hr of incubation, although numerous fluorescent punctae appeared (*Figure 6a,c,e*). Because TOR is the major negative regulator of autophagy (*Pu et al., 2017*), the punctae are presumably autophagic structures resulting from the TOR inhibition by the chemicals. Untreated *eva1* cells contained numerous small vacuoles and conspicuous TVSs (*Figure 6b*); however, by 2 hr treatment with AZD-8055, these structures were reduced in appearance (*Figure 6d*). By 4 hr treatment, the *eva1* cells were indistinguishable from WT, including the appearance of the small punctae (*Figure 6e,f*). These results were mirrored by Torin2 treatment: presumably unfused vacuoles and TVSs were no longer present in the *eva1* cells by 2 hr of 1 μM Torin2 treatment (*Figure 6—figure supplement 1*). This result is in line with the higher in vitro TOR inhibitory activity of Torin2 compared to AZD-8055 (*Liu et al., 2011*; *Chresta et al., 2010*). In addition to the effects of temporal treatment on vacuolar phenotype (*Figure 6g*), we tested the effects of chronical inhibition of TOR. We found *ipms1* primary root elongation was promoted by lower concentrations (0.1 and 0.2 μM), but inhibited by higher concentrations (0.4, 0.6 and 1.0 μM) of AZD-8055 (*Figure 6—figure supplement 2*), suggesting moderate TOR inhibition led to optimized plant growth of *ipms1*. Similarly, a low concentration of wortmannin caused minimal but significant promotion of *ipms1* root elongation (*Figure 6—figure supplement 3a–c*). By comparison, promotion of root elongation was not observed upon Lat B treatment (*Figure 6—figure supplement 3d–f*). Together these results indicate that both subcellular and growth defects of *ipms1* are associated with up-regulated TOR and are suppressed by TOR inhibition.

We next sought to confirm these results by testing the activation status of TOR in *ipms1*. Based on the evidence that TOR inhibition rescued the *ipms1* subcellular phenotypes, we predicted to find an increased level of TOR activity in *ipms1* compared to WT. S6K is a conserved substrate of TOR protein kinase and its phosphorylation status has been adopted as an indicator of TOR activity in plants (*Pfeiffer et al., 2016*; *Wang et al., 2018*; *Dong et al., 2017*; *Xiong and Sheen, 2012*). Indeed, immunoblot analyses with specific antisera for either phosphorylated or total S6K (*Pfeiffer et al., 2016*; *Wang et al., 2018*; *Dong et al., 2017*; *Xiong and Sheen, 2012*) revealed increased levels of TOR-phosphorylated S6K in *eva1* and *ipms1-4* compared to WT, despite similar levels of total S6K in three genotypes (*Figure 6h,i*). These data supports that TOR signaling is up-regulated in the *ipms1* background. To validate this conclusion, we monitored DNA synthesis in root tips because a stimulated TOR signaling promotes cell proliferation in the root apical meristem, which can be detected by EdU staining of newly synthesized DNA (*Li et al., 2017*; *Dong et al., 2017*; *Xiong et al., 2013*). Consistent with our hypothesis, the EdU staining displayed enhanced labeling in the root apical meristem of *ipms1-4* and *ipms1-5* compared to WT (*Figure 6j,k*). This result was supported by propidium iodide staining and morphometric analyses of root tips showing increased cell numbers in the root apical meristem of *eva1*, *ipms1-4* and *ipms1-5* compared to WT (*Figure 3—figure supplement 2c*).

Taken together, the results indicating suppression of vacuole phenotypes by TOR inhibition, increased levels of S6K phosphorylation and root apical meristem activity (i.e., increased DNA synthesis and cell number) in the *ipms1* mutants support the hypothesis that TOR signaling is up-regulated in the *IPMS1* loss-of-function mutants.

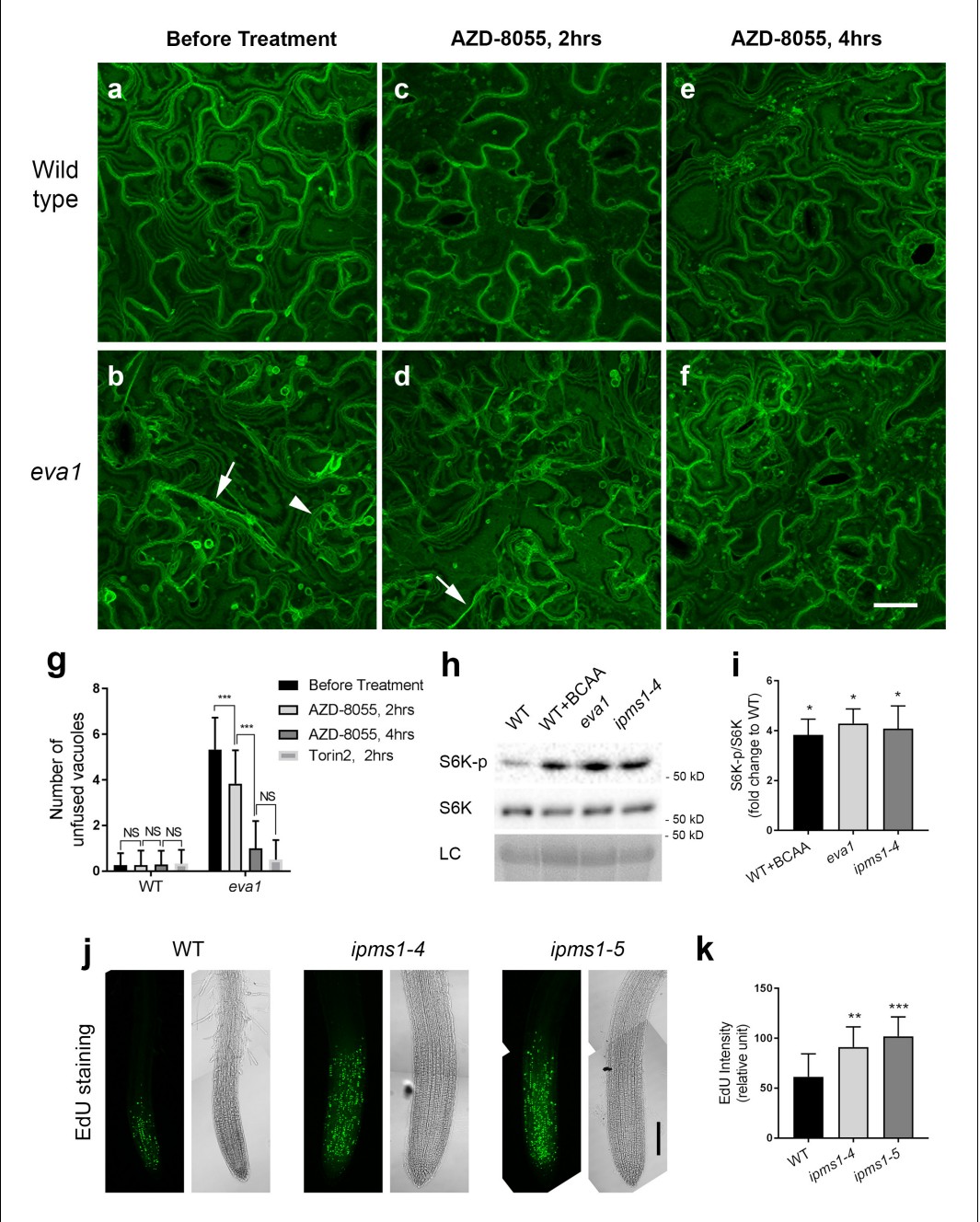

**Figure 6.** Vacuolar mutant phenotypes of *eva1* are correlated with up-regulated TOR activity. (**a – f**) TOR inhibitor AZD-8088 treatment rescues vacuolar mutant phenotypes of *eva1*. Confocal images were acquired before (**a and b**), after 2 hr (**c and d**), and after 4 hr (**e and f**) 5 μM AZD-8055 treatment of wild type and *eva1* mutant. Arrowhead indicates presumably unfused vacuolar structures and arrows point to enhanced TVSs. All the images are Z-stack maximal projections. Scale bar, 25 μm. (**g**), Quantification of presumably unfused vacuoles in *eva1* before and after TOR inhibitor treatment. Values are mean ± SD. The asterisks indicate significant differences (n = 30 cells for each treatment; ***p≤0.001, NS, p>0.05 and not significant, unpaired *t* test). (**h**) Immunoblotting detected phosphorylation of S6K by TOR, using specific antisera against S6K-phosphorylated and S6K. LC, loading control with Ponceau S staining. (**i**) S6K phosphorylation status calculated by the ratio of S6K-p/S6K in fold change compared to wild type. n = 3 and values are mean ± SEM (***p≤0.05, unpaired *t* test). (**j**) EdU staining detected root meristem activity of 10 day old seedlings. For each genotype, an image of green pseudocolor shows EdU-stained newly synthesized DNA and a bright-field image shows structure of root tip. Scale bar, 100 μm. (**k**)

*Figure 6 continued on next page*

*Figure 6 continued*

Quantification of EdU fluorescence intensity. Values are mean ± SD. The asterisks indicate significant differences compared to wild type (n = 9 for each genotype; **p≤0.01, ***p≤0.001, unpaired *t* test).

The online version of this article includes the following figure supplement(s) for figure 6:

**Figure supplement 1.** Another TOR inhibitor Torin2 exerts more potent effects on the vacuolar phenotypes of *eva1*.

**Figure supplement 2.** The effect of TOR inhibitor AZD-8055 on *ipms1* primary root elongation is dose-dependent.

**Figure supplement 3.** Effects of PI3K/TOR dual inhibitor wortmannin and F-actin depolymerizer Lat B on *ipms1* primary root elongation.

## Over-accumulation of BCAAs alters the subcellular organization of the actin cytoskeleton and endomembranes

Furthermore, we aimed to test a role of TOR signaling and its specificity in the verified BCAA over-accumulation-induced phenotypes. To do so, we utilized an estradiol-inducible TOR mutant (*tor-es*) (*Xiong and Sheen, 2012*) and a loss-of-function mutant of *AtRAPTOR1B* (*raptor1b*, SALK_022096) (*Salem et al., 2017*), a locus encoding the functional TORC1 component RAPTOR in Arabidopsis (*Salem et al., 2018*; *Anderson et al., 2005*). Before silencing induction, similar to WT (*Figure 7a,e*), *tor-es* seedlings grown on BCAA-supplemented medium showed induced F-actin bundling compared to *tor-es* grown on normal medium (*Figure 7b,f*). After induction of TOR silencing, *tor-es* grown on either medium exhibited similarly low levels of bundling (*Figure 7c,g*). These results confirm a functional dependence of TOR signaling and the actin cytoskeleton phenotype due to misregulated TOR. By contrast, in *raptor1b* BCAA feeding led to F-actin bundling (*Figure 7d,h*). Together, these results not only indicate that reorganization of F-actin induced by over-accumulation of BCAAs is dependent on functional TOR but also underlie a cause of the subcellular phenotype linked to BCAA on TOR signaling components other than RAPTOR.

Next, we aimed to test the generality of the connection between over-accumulation of BCAAs, morphological alteration of cellular structures and functional TOR signaling. To do so, we used a variety of previously characterized BCAA mutants (*Xing and Last, 2017*), combined with BCAA feeding. For example, *ipms1-1*[D] was chosen because it has a modest Val decrease and Leu increase; *ahass1-1* has a small Val increase; *ahass2-7* has decreased Val and Leu; *omr1-11*[D] has a > 140 fold Ile increase compared to WT. Confocal microscopy analyses of cotyledon epidermal cells revealed that the organization of F-actin in *ipms1-1*[D] and *ahass1-1* mutants resembled that of WT (*Figure 7a,i and j*). By contrast, enhanced actin bundling was observed following BCAA feeding (1 mM Val, Leu and Ile) and in the *ipms1-5* and *omr1-11*[D] mutants (*Figure 7e,k and l*). Interestingly also, we found that the mutants showed reorganization of F-actin and remodeling of the ER network. Specifically, mutants with moderate increase or decrease in BCAAs showed ER morphology similar to WT (*Figure 7—figure supplement 1a–d*), while WT grown with BCAA supplementation and mutants that over-accumulate BCAAs showed compromised ER organization with longer and thicker ER strands compared to WT (*Figure 7—figure supplement 1e–g*). The striking phenotype of enhanced ER strands in *omr1-11*[D] was recovered by a 2 hr Torin2 treatment (*Figure 7—figure supplement 1h*). In addition to bundling of F-actin and enhancement of ER strands, supplementation of BCAAs also induced the formation of prominent TVSs (*Figure 7—figure supplement 2*). Together, these results support a general correlation between over-accumulation of BCAAs and distorted actin cytoskeleton and endomembranes.

## Discussion

In eukaryotic cells, the TOR kinase coordinates cell growth and metabolism with nutrient sensing (*Saxton and Sabatini, 2017*; *Kim and Guan, 2019*). In mammalian cells, the availability of BCAAs and other amino acids – as well as glucose and mammalian growth factors – regulates TOR signaling, which generally promotes growth via several downstream cellular processes, including mRNA translation, metabolism of nucleotides, sugar and lipids, protein turnover and cytoskeletal reorganization (*Saxton and Sabatini, 2017*; *Kim and Guan, 2019*). Prior to this work, in plant cells a functional connection between glucose availability and TOR activation had been established (*Shi et al., 2018*).

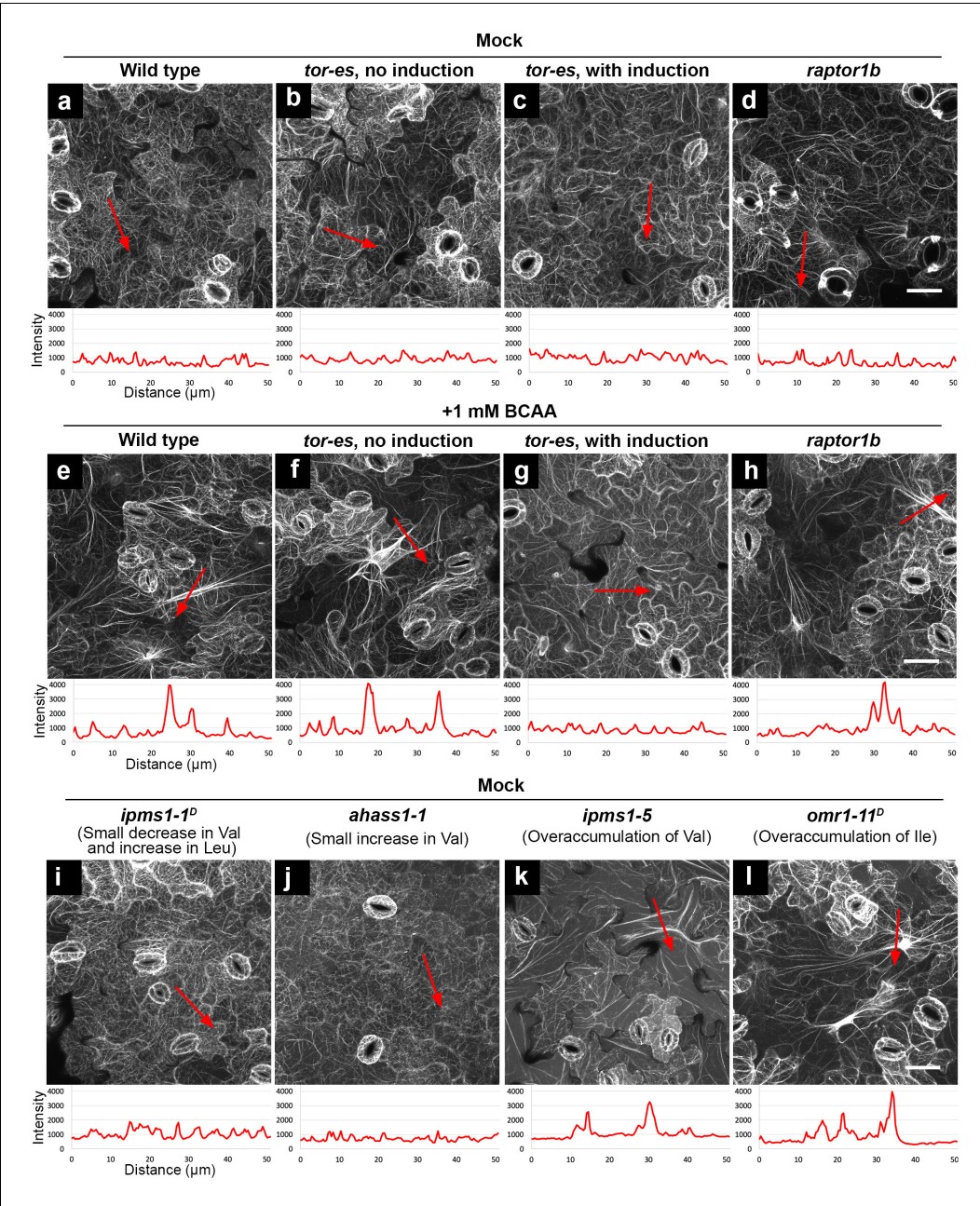

**Figure 7.** Feeding of exogenous BCAAs and over-a ccumulation of endogenous BCAAs induce actin bundling, which is dependent on functional TOR but not RAPTOR. Organization of actin cytoskeleton presented by confocal images of cotyledon epidermal cells expressing F-actin marker YFP-ABD2. Higher fluorescence intensity of the actin marker suggests more bundling of actin filaments. Using ImageJ, we drew a 50 μm red arrowed line to detect the pixel fluorescence intensity beneath such a line. In each image, the red arrowed line is positioned where we detected the highest fluorescence intensity using non-saturating imaging settings. A chart beneath the image presents plotted fluorescence intensity along the red arrowed line. Without actin bundling, fine actin filaments have fluorescence intensity about 1000 (relative unit). In contrast, induced actin bundling show fluorescence intensity peaks of 3000–4000 (relative unit). (a – d) Without feeding of BCAAs (mock), wild type, *tor-es* with or without silencing, and *raptor1b* did not show induced actin bundling. (e – h) Feeding of 1 mM BCAAs induced striking actin bundling in wild type, *tor-es* without gene silencing and *raptor1b* (e, f, h), but not in *tor-es* with induction of TOR silencing (g). (i – l), Without feeding of BCAAs (mock), mutants with small changes of BCAAs did not show induced actin bundling (i and j); mutants with over-accumulation of endogenous BCAAs showed induced actin bundling (k and l). All the images are Z-stack maximal projections. Scale bars, 50 μm.

*Figure 7 continued on next page*

*Figure 7 continued*
The online version of this article includes the following figure supplement(s) for figure 7:

**Figure supplement 1.** Exogenous feeding of BCAAs or over-accumulation of BCAAs due to mutations of the BCAA biosynthetic pathway alters morphology of the ER network to form enhanced ER strands.

**Figure supplement 2.** Exogenous feeding of BCAAs affects morphology of the lytic vacuole.

Besides, a recent research found that mutation of IPMS1 leads to alteration of Leu biosynthesis and reduced sensitivity to TOR inhibition (*Schaufelberger et al., 2019*), but a role for BCAA homeostasis in TOR signaling was yet unknown. Furthermore, it was still unclear whether TOR signaling is connected to the morphogenesis and remodeling of subcellular structures, other than autophagic bodies. We demonstrated that TOR signaling senses BCAA homeostasis and modulates the organization of subcellular structures, including actin cytoskeleton and endomembranes. Indeed, we provide evidence that over-accumulation of BCAAs up-regulates TOR signaling, inducing actin bundling with formation of aberrant endomembrane structures and compromising overall growth (*Figure 8*).

## Endogenous BCAAs influence TOR signaling in plant cells

In yeast and mammalian cells, TOR signaling is modulated directly by changes in amino acid levels, especially BCAAs, and indirectly by glucose levels and growth factors (*Saxton and Sabatini, 2017*; *Kim and Guan, 2019*). In plants, TOR-mediated meristem activation by sugar and light signals has been established (*Li et al., 2017*; *Chen et al., 2018*; *Pfeiffer et al., 2016*; *Xiong et al., 2013*). In addition, a recent research suggested a connection between IPMS1-regulated Leu biosynthesis and TOR signaling (*Schaufelberger et al., 2019*). Specifically, this work identified another *IPMS1* mutant allele *rol17* with shorter primary root, and its primary root elongation is hyposensitive to TOR inhibitor treatment (*Schaufelberger et al., 2019*). Considering that plants produce BCAAs de novo and their biosynthesis is stringently controlled (*Xing and Last, 2017*), it is necessary to explore a general connection between TOR signaling and amino acid homeostasis.

In this study, using a series mutants of BCAA biosynthesis, exogenous feeding of BCAAs and mutants of TOR signaling, we provide evidence for a functional correlation between BCAA accumulation and TOR activity in plant cells. Indeed, we found that in early stage of growth (i.e., 10 days) BCAA over-accumulating mutants showed up-regulation of TOR, a phenotype that was recreated by BCAA feeding. Furthermore, a reduction in TOR signaling activation via dual and specific TOR inhibitors restored the actin and endomembrane phenotypes. Our results establish that plant TOR senses over-accumulation of BCAAs, and that up-regulation of TOR signaling alters the organization of actin cytoskeleton and associated endomembranes and controls plant growth. Our results indicate that plant TOR signaling and TOR-dependent growth regulation are highly responsive to BCAA availability, at least in early stages of growth, underscoring a previously unappreciated but significant role of these amino acids in TOR biology.

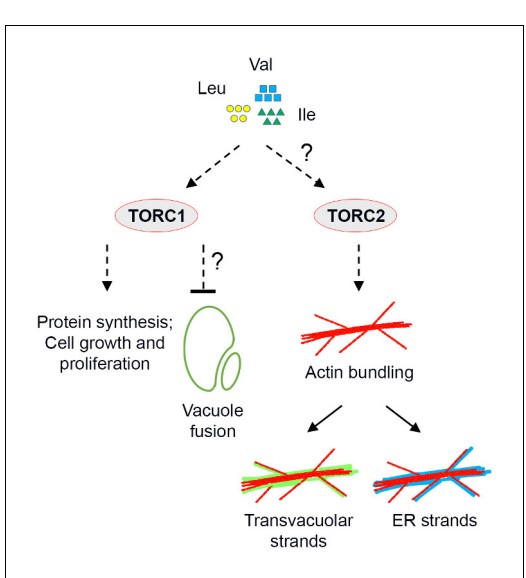

**Figure 8.** Working model of TOR-regulated subcellular processes. Over-accumulation of BCAA Val, Leu and Ile stimulates TOR signaling. Except for the established downstream processes such as protein synthesis and cell proliferation, vacuole fusion and actin reorganization are indicated also regulated by TOR signaling, but the underlying mechanisms are unclear. Reorganization of the actin cytoskeleton is independent of TORC1, and prominent *trans*-vacuolar strands and ER strands are subsequently formed due to the strong interactions between the endomembranes and the F-actin in plant cells.

## TOR signaling controls actin organization in plant cells

In yeast and mammalian cells, a reorganization of actin cytoskeleton was the first identified downstream effect specific to TORC2, which was also found to be independent of TORC1 (*Sarbassov et al., 2004*; *Jacinto et al., 2004*; *Schmidt et al., 1996*). The yeast and mammalian TORC2 phosphorylates other protein kinases, which signal to the Rho-coordinated cytoskeletal signaling (*Xie et al., 2018*; *Roelants et al., 2017*). Recently, it has been demonstrated that Arabidopsis ROP2 interacts with TOR and mediates an auxin-to-TOR signaling (*Li et al., 2017*; *Schepetilnikov et al., 2017*), suggesting an interplay among TOR, ROPs and the cytoskeleton in plant cells. In terms of TORC2 activation, in mammalian cells, Rictor and mTOR, but not Raptor, are required for PKC phosphorylation and actin organization (*Sarbassov et al., 2004*), and nutrient repletion after starvation stimulates actin organization (*Jacinto et al., 2004*). Nevertheless, much less is known about the activation mechanism of mTORC2 than that of mTORC1. Therefore, how BCAAs activate a potential TORC2 in plant cells is still an open question (*Figure 8*). The evidence provided in our work for promoted actin bundling phenotype in mutants with BCAA over-accumulation or upon BCAA supplementation to WT provides direct support for a functional interaction between TOR and the actin cytoskeleton. Furthermore, our evidence that the BCAA-induced actin reorganization relies on functional TOR but not RAPTOR, which defines TORC1, extends these conclusions at a mechanistic level about the exclusive functions of TOR interactors and specific downstream processes.

## Plant endomembrane homeostasis is correlated with TOR activity

In this work, we have shown that an over-accumulation of BCAAs affects the morphology of the tonoplast, ER and Golgi distribution. Our results also showed that attenuation of TOR signaling in BCAA over-accumulating mutants leads to a restoration of the defective actin cytoskeleton and endomembrane morphology. We propose therefore that in plant cells a stringent relationship exists between endomembrane organization and TOR-signaling, which occurs via a functional connection of TOR with the actin cytoskeleton. Depending on nutrient status, TORC1 regulates the size and number of yeast vacuoles and mammalian lysosomes (*Yu et al., 2010*; *Michaillat et al., 2012*). In light of the conserved functions of nutrient storage and turnover that are shared by plant and yeast vacuoles (*Zhang et al., 2014*), a TORC1-dependent control of the vacuolar homeostasis may be conserved in plant cells. An auxin-dependent actin remodeling has been also recently invoked in controlling vacuole occupancy in the plant cell (*Löfke et al., 2015*; *Scheuring et al., 2016*). Additionally, a recent work reported regulation of vacuole expansion by extracellular LRX (*Dünser et al., 2019*), whose mutation resulted in root hair defects that can be suppressed by mutation of IPMS1 (*Schaufelberger et al., 2019*). In this work, we provided evidence for a remodeling of actin through TORC1-independent TOR signaling. Therefore, we propose that TOR activity is involved in morphogenesis of the central vacuole, but this may occur through mechanisms that are not conserved (*Figure 8*).

## Plant growth and TOR signaling

BCAAs are crucial nutrients that humans and other animals must obtain from diets. Significant deficits exist in the amino acid composition of plant feed sources for livestock (*Boisen et al., 2000*). Moreover, BCAAs serve as human dietary supplements, because they potently promote protein synthesis through TOR activity (*Shimomura et al., 2004*). Fortification of crops with BCAAs is therefore desirable to improve plant nutritional content. However, studies in Arabidopsis reported that BCAA over-accumulation due to exogenous feeding or genetic manipulation of biosynthetic or catabolic pathways resulted in various defects in plant growth and development (*Xing and Last, 2017*; *Peng et al., 2015*; *Field et al., 2006*; *Imhof et al., 2014*). In this study, we established a functional connection between BCAA over-accumulation and plant growth inhibition, which is likely linked to alteration of TOR signaling and disruption of subcellular structures. Despite an overall retardation of early plant development compared to WT, *ipms1* seedlings exhibited up-regulation of the growth-promoting TOR signaling in concert with locally promoted growth, such as larger and thicker cotyledons and higher activity of root apical meristem. At the subcellular level, we showed that BCAA over-accumulation via exogenous feeding or genetic mutations caused severe remodeling of the actin cytoskeleton and endomembranes, which underlies certain growth defects such as delayed

formation of root hairs and reduced abundance of lateral roots (*Stefano et al., 2012*; *Ketelaar et al., 2004*; *Deeks et al., 2005*). In addition, *ipms1* mutants show increase in leaf thickness and alteration of chloroplast ultrastructure, which is similar to a knockdown mutant of isopropylmalate isomerase (IPMI), the enzyme following IPMS for Leu biosynthesis (*Imhof et al., 2014*), suggesting an association of BCAA accumulation with defective chloroplast development. Together, our results indicate that an inconsistency between nutrient status and the activity of metabolic signaling is detrimental to plant growth and development. By adding new insights into the fundamental understanding of plant growth control by metabolic signaling, our study supports that developmentally controlled manipulation of TOR signaling may be required for successful engineering of crops with improved yield and nutritional values.

# Materials and methods

**Key resources table**

| Reagent type (species) or resource | Designation | Source or reference | Identifiers | Additional information |
|---|---|---|---|---|
| Gene (*Arabidopsis thaliana*) | *AtIPMS1* | | TAIR: AT1G18500 | |
| Gene (*Arabidopsis thaliana*) | *AtOMR1* | | TAIR: AT3G10050 | |
| Gene (*Arabidopsis thaliana*) | *AtAHASS1* | | TAIR: AT2G31810 | |
| Gene (*Arabidopsis thaliana*) | *AtAHASS2* | | TAIR: AT5G16290 | |
| Gene (*Arabidopsis thaliana*) | *AtTOR* | | TAIR: AT1G50030 | |
| Gene (*Arabidopsis thaliana*) | *AtRaptor1B* | | TAIR: AT3G08850 | |
| Genetic reagent (*Arabidopsis thaliana*) | *eva1* | this paper | | EMS line with a mutation of *IPMS1* |
| Genetic reagent (*Arabidopsis thaliana*) | *ipms1-4* | *Xing and Last, 2017* | SALK_101771 | |
| Genetic reagent (*Arabidopsis thaliana*) | *ipms1-5* | *Xing and Last, 2017* | WiscDsLoxHs221_05F | |
| Genetic reagent (*Arabidopsis thaliana*) | *tfl111 (ipms1-1$^D$)* | *Xing and Last, 2017* | TAIR: CS69734 | |
| Genetic reagent (*Arabidopsis thaliana*) | *tfl102 (ipms1-1$^D$)* | *Xing and Last, 2017* | TAIR: CS69733 | |
| Genetic reagent (*Arabidopsis thaliana*) | *ahass1-1* | *Xing and Last, 2017* | SALK_096207 | |
| Genetic reagent (*Arabidopsis thaliana*) | *ahass2-7* | *Xing and Last, 2017* | WiscDsLoxHs009_02G | |
| Genetic reagent (*Arabidopsis thaliana*) | *ahass2-1$^D$* | *Xing and Last, 2017* | TAIR: CS69724 | |
| Genetic reagent (*Arabidopsis thaliana*) | *omr1-11$^D$* | *Xing and Last, 2017* | TAIR: CS69720 | |
| Genetic reagent (*Arabidopsis thaliana*) | *tor-es* | *Xiong and Sheen, 2012* | TAIR: CS69829 | |
| Genetic reagent (*Arabidopsis thaliana*) | *raptor1b* | *Salem et al., 2017* | SALK_022096 | |
| Antibody | Anti-S6K (Rabbit polyclonal) | Agrisera | AS12 1855 | Western blotting (1:1000 dilution) |
| Antibody | Anti-S6K-phosphorylated (Rabbit polyclonal) | Abcam | ab207399 | Western blotting (1:1000 dilution) |

*Continued on next page*

*Continued*

| Reagent type (species) or resource | Designation | Source or reference | Identifiers | Additional information |
|---|---|---|---|---|
| Antibody | HRP conjugated anti-rabbit (Goat polyclonal) | Sigma-Aldrich | A0545 | Western blotting (1:10000 dilution) |
| Commercial assay or kit | Click-iT EdU Alexa Fluor 488 Imaging Kit | Invitrogen | C10337 | |

## Plant materials and growth conditions

An EMS mutant line first identified in a screen for vacuolar phenotypes (*Avila et al., 2003*) was crossed with wild type (Col-0) for three times to obtain a progeny with consistently inherited vacuolar and growth phenotypes, which was designated as *eva1*.

Except for chronical treatments with specified chemicals, Arabidopsis seeds were stratified and grown on medium containing half-strength Linsmaier and Skoog nutrients (½ LS; Caisson Labs, LSP03), 1% sucrose and 0.4% phytagel (Sigma-Aldrich, P8169) in chambers configured with 21°C and 16 hr light: 8 hr dark cycle.

To examine the effect of latrunculin B (Lat B) on root elongation (*Figure 4—figure supplement 4*), wild type (Col-0), *ipms1-4* and *ipms1-5* lines germinated and grew on horizontally staged Petri dishes containing Arabidopsis growth medium (½ LS, 1% sucrose and 0.4% phytagel). 10 day old seedlings were transplanted to Petri dishes containing ½ LS, 1% sucrose and 1% Agar (Acumedia, 7558A) medium containing DMSO or 50 nM Lat B or 100 nM Lat B. Photographs were acquired immediately after the transplant and the Petri dishes were vertically staged in a Percival chamber. Photographs were also acquired 8 days after the transplant.

In another pharmaceutical examination using AZD-8055, wortmannin and Lat B (*Figure 6—figure supplement 2*; *Figure 6—figure supplement 3*), wild type (Col-0), *eva1*, *ipms1-4* and *ipms1-5* lines germinated and grew on vertically staged Petri dishes containing Arabidopsis growth medium containing specific chemical inhibitors.

Exogenous feeding of 1 mM BCAA was performed by stratification and germination of seeds on ½ LS, 1% sucrose and 1% Agar medium containing 1 mM equal concentrations of Ile, Val and Leu. L-Isoleucine (Sigma-Aldrich, I2752), L-Valine (Sigma-Aldrich, V0500) and L-Leucine (Sigma-Aldrich, L8000) were dissolved in water to prepare 1 M stock solutions, which were filtered by Millex-GS 0.22 µm filter units (Millipore, SLGS033SS).

## Confocal microscopy

A Zeiss LSM 510 META and a Nikon A1Rsi laser scanning confocal microscope were used for imaging. Acquired images were handled by NIS-Elements Advanced Research (Nikon), ZEN (Zeiss) and Fiji (ImageJ) (*Schindelin et al., 2012*). The fluorescent protein fusions used in this study are GFP-δ TIP (*Cutler et al., 2000*), ERYK (*Nelson et al., 2007*), YFP-ABD2 (*Sheahan et al., 2004*), GFP-CASP (*Renna et al., 2005*), SEC-RFP (*Faso et al., 2009*) and γTIP-YFP (*Nelson et al., 2007*). Transformation of Arabidopsis plants were conducted using floral dip method (*Clough and Bent, 1998*).

## Quantitative analysis of ER morphology and actin cytoskeletal organization

Image acquisition and further evaluation of the ER cisternae was conducted using a previously described method (*Cao et al., 2016*) that measures the occupancy of ER area in a region of interest. Analyses of the actin cytoskeletal organization were performed following a previously described procedure (*Lu and Day, 2017*). Briefly, Z-stack images with 0.5 µm intervals were acquired to cover the whole epidermal cell. The Z-stack series were converted to maximal projection images using NIS-Elements Advanced Research (Nikon) and Fiji (ImageJ) (*Schindelin et al., 2012*). Utilizing two ImageJ macros that were previously generated (*Lu and Day, 2017*), skewness was measured to present the distribution of YFP-ABD2 fluorescence intensity and occupancy was measured for the density of skeletonized YFP-ABD2 fluorescence signal.

## Chemical stocks and treatments

All temporal chemical treatments were performed using 10 day old seedlings. Each of the following chemicals was first dissolved in DMSO to prepare a stock solution, and then diluted in Arabidopsis growth medium (½ LS and 1% sucrose) to reach the specific working concentration. 33 µM Wortmannin (Sigma-Aldrich, W1628) and 100 µM LY294002 (MedChemExpress, HY-10108) were used to treat seedlings for 2 hr. Latrunculin B (Sigma-Aldrich, L5288) and Oryzalin (Chem Service Inc, N-12729) were diluted to 25 µM and 40 µM, respectively, for 2 hr treatments. For TOR inhibition, seedlings were incubated with 5 µM AZD-8055 (MedChemExpress, HY-10422) or 1 µM Torin2 (MedChemExpress, HY-13002) for 2 or 4 hr as the figure legends indicated. 10 µM solution of β-estradiol (Sigma-Aldrich, E8875) was used to induce gene silencing.

## Amino acid extraction and LC-MS/MS analysis

Plants used for amino acid extraction were grown under standard conditions for 10 or 20 days. The aerial tissue (fresh weight around 10 mg) was harvested into a 2 mL tube with two 3 mm steel beads and flash frozen in liquid N2. Tissue was either used immediately or stored at −80°C until extraction. Tissue was pulverized using a mixer mill (Retsch Mill, MM400) for 1 min at 30 times per second. Amino acids were extracted as previously reported (*Xing and Last, 2017*; *Angelovici et al., 2013*). Briefly, an amino acid extraction buffer was prepared with ~2 µM heavy labeled amino acids standards (13C, 15N, Sigma-Aldrich), 10 µM 1,4-dithiothreitol (DTT, Sigma-Aldrich), and 10 mM perfluoroheptanoic acid (PFHA, Sigma-Aldrich). To the ground tissue, 350 µL of extraction buffer was added, vortexed for 10 s and heated at 90°C for 10 min. Tubes were cooled on ice for 5 min and centrifuged for 10 min at 4°C at 13,000 × g. The supernatant was applied to a low-binding hydrophilic 0.2 µm centrifugal polytetrafluoroethylene (PTFE) filter (Millipore, UFC30LG25) and centrifuged for 5 min at 3,500 × g. 150 µL flow through was transferred to 2 mL glass vials with glass insert for LC-MS analysis.

Amino acid detection and quantification by LC-MS/MS was performed as previously reported (*Xing and Last, 2017*; *Angelovici et al., 2013*). Briefly, a dilution series (12.2 nM to 250 µM) of each individual amino acid standard was made containing the same concentration of the heavy standards as was in the amino acid extraction buffer. Samples were injected into a Quattro micro API LC/MS/MS (Waters) equipped with an Acquity UHPLC HSS T3 1.8 µm column (Waters) using a three-function method. The 13 min LC method with solvent A (10 mM PFHA) and solvent B (acetonitrile) at a flow rate of 0.3 mL/minute is provided in *Supplementary file 1*. Amino acids were quantified by comparison to their standard curves using QuanLynx.

## EdU staining

EdU (5-ethynyl-2′-deoxyuridine) staining of root apical meristem was performed using Click-iT EdU Alexa Fluor 488 Imaging Kit (Invitrogen, C10337), following a protocol that was adapted for plant tissues (*Kotogány et al., 2010*). Labeling was performed by incubating 10 day old Arabidopsis seedlings in 10 µM EdU in Arabidopsis growth medium (½ LS and 1% sucrose) for 30 min in a Percival chamber. All samples were then incubated with a fixation buffer (4% formaldehyde, 0.1% Triton X-100, 1 × PBS) for 30 min. All samples were washed for three times, 10 min each, with 1 × PBS after fixation. The EdU detection was conducted by 30 min incubation in dark with the Click-iT cocktail, which was prepared according to the manual of Click-iT EdU Alexa Fluor 488 Imaging Kit. Each sample was immediately washed for three times, 10 min each, with 1 × PBS before imaging.

## PI staining of root tip and measurement

PI (propidium iodide) staining was conducted by 3 min incubation of Arabidopsis seedlings in propidium iodide (Invitrogen, P3566) diluted to 1 µg/mL using Arabidopsis growth medium (½ LS and 1% sucrose). After staining, all samples were immediately washed for 1 min and then subjected to imaging. Confocal images of propidium iodide stained root tips were analyzed using Cell-O-Tape (*French et al., 2012*), which is a plugin of ImageJ that automatically segments three zones in a root tip (the meristem, the transition zone and the mature zone) by comparing the lengths of adjacent cells in the same cortical layer. Adjacent cells with significant increase in cell length belong to the transition zone. Cells before and after the transition zone are categorized as cells in the meristem

and the mature zone respectively. The program records the length of each cell and the cell number in each zone.

## Protein preparation and immunoblotting

To detect the phosphorylation status of S6K, 50 mg plant aerial tissue was used for protein extraction using 1.5 mL extraction buffer of 1 × PBS, pH 7.4, containing 250 mM sucrose, Protease Inhibitor Cocktail (Sigma-Aldrich, P9599) and PhosSTOP phosphatase inhibitor (Roche, 4906845001). Three times of centrifugation, 1 k × g for 5 min, 14 k × g for 5 min and 135 k × g for 30 min, were conducted to separate the soluble proteins. The supernatant from the last centrifugation was separated, concentrated to 200 μL using an Amicon Ultra centrifugal unit (Millipore, UFC501024), and then mixed with 40 μL 6 × Laemmli buffer. Proteins were denatured by incubation at 95℃ for 10 min. Protein samples were separated on 15% SDS-PAGE with 8M urea and blotted to PVDF membranes (Bio-Rad, 1620177). Blots were blocked with 5% milk for 1 hr at room temperature. Blots were incubated with primary antibodies of either anti-S6K (Agrisera, AS12 1855) or anti-S6K-phosphorylated (Abcam, ab207399) overnight at 4℃ and subsequently with secondary HRP conjugated goat anti-rabbit antibody (Sigma-Aldrich, A0545) for 1 hr at room temperature.

## Extraction and measurement of anthocyanins

The aerial parts of 10 day old Arabidopsis seedlings were collected, and then lyophilized and measured for dry weight. Total anthocyanins were extracted using 1 μL extraction buffer (50% methanol containing 3% formic acid) per 50 μg dry weight. After overnight incubation with extraction buffer at room temperature, the supernatant was collected and measured absorbance of 532 nm.

## TEM and measurement of leaf thickness

The electron microscopic imaging of the endomembrane structures and chloroplasts were performed following an established protocol (*Kim et al., 2018*). In brief, 1 mm ×1 mm pieces of cotyledon samples were cut and fixed in TEM fixative buffer (2.5% paraformaldehyde and 2.5% glutaraldehyde in 0.1 M cacodylate buffer, pH 7.4) with vacuum infiltration. The fixed samples were stained with 1% osmium tetroxide overnight at 4℃. After series of dehydration with acetone, the samples were infiltrated and embedded in Spurr's Resin. Sections with 50 nm thickness were cut and mounted on the copper grid and 10 well slides. For TEM, the grids were post-stained in 2% uranylacetate for 30 min and then treated with 1% lead citrate for 15 min. JEOL 100CX TEM (JEOL USA) was used to observe the ultrastructure of cotyledon.

The thickness of cotyledons was measured as previously described (*Weraduwage et al., 2016*). Briefly, 2 mm ×1 mm samples cut from the center of the cotyledons were fixed in fixative buffer (4% paraformaldehyde and 0.5% glutaraldehyde in 1 × PBS, pH 7.4) with vacuum infiltration. The fixed samples were stained with 1% osmium tetroxide overnight at 4℃. After series of dehydration with acetone, the samples were infiltrated and embedded in Spurr's Resin. Sections with 500 nm thickness were cut and mounted on the copper grid and 10 well slides. For leaf thickness analysis, the sections were stained with 1% toluidine blue for 1 min and washed with running water. Images were taken using Axio Imager M2 (Zeiss), and measurement of leaf thickness was performed using AxioVision SE64 Rel. 4.9.1 (Zeiss) software. Three biological samples with three technical replicates were used to measure leaf thickness.

## Acknowledgements

We thank Dr. Natasha Raikhel for the kind donation of the EMS mutant line. We thank the members of the Brandizzi laboratory for critical inputs and Dr. Xiayan Liu for helpful discussions. We thank Drs. Brad Day and Yi-Ju Lu for assistance of quantitative evaluation of the actin cytoskeleton. We thank the MSU Mass Spectrometry and Metabolomics Core Facility for assistance in amino acid analysis. We thank the MSU Center for Advanced Microscopy and Dr. Melinda Frame for assistance in confocal microscopy. We acknowledge support by the Chemical Sciences, Geosciences and Biosciences Division, Office of Basic Energy Sciences, Office of Science, US Department of Energy (award number DE-FG02-91ER20021) for infrastructure, and the National Science Foundation (MCB1727362), the DOE Great Lakes Bioenergy Research Center (DOE BER Office of Science

DE-SC0018409) and AgBioResearch (MICL02598) to FB and the National Science Foundation post-doctoral fellowship (NPGI-1811055) to CAS.

## Additional information

### Funding

| Funder | Grant reference number | Author |
|---|---|---|
| National Science Foundation | MCB1727362 | Federica Brandizzi |
| National Institute of Food and Agriculture | MICL02598 | Federica Brandizzi |
| National Science Foundation | NPGI-1811055 | Craig A Schenck |
| U.S. Department of Energy | DE-FG02-91ER20021 | Federica Brandizzi |
| U.S. Department of Energy | DE-SC0018409 | Federica Brandizzi |

The funders had no role in study design, data collection and interpretation, or the decision to submit the work for publication.

### Author contributions

Pengfei Cao, Conceptualization, Formal analysis, Investigation, Visualization, Methodology; Sang-Jin Kim, Jie Wang, Formal analysis, Investigation, Visualization, Methodology; Anqi Xing, Conceptualization, Resources, Methodology; Craig A Schenck, Formal analysis, Funding acquisition, Investigation, Methodology; Lu Liu, Formal analysis, Validation, Investigation; Nan Jiang, Formal analysis, Investigation, Methodology; Robert L Last, Conceptualization, Supervision; Federica Brandizzi, Conceptualization, Supervision, Funding acquisition, Methodology, Project administration

### Author ORCIDs

Pengfei Cao (iD) https://orcid.org/0000-0001-6998-9302
Craig A Schenck (iD) http://orcid.org/0000-0002-5711-7213
Robert L Last (iD) http://orcid.org/0000-0001-6974-9587
Federica Brandizzi (iD) https://orcid.org/0000-0003-0580-8888

### Decision letter and Author response

Decision letter https://doi.org/10.7554/eLife.50747.sa1
Author response https://doi.org/10.7554/eLife.50747.sa2

## Additional files

### Supplementary files

- Supplementary file 1. LC gradient used for amino acid separation.
- Supplementary file 2. Primers used in this study.
- Transparent reporting form

### Data availability

All data generated or analyzed in this study have been included in the manuscript and the supporting files.

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
