## [Decision Letter]

**Acceptance summary:**

We are excited about this work, because the authors elegantly utilized genetic and cell biological approaches to reveal how plant growth is linked to nutrition sensing. The authors’ forward genetic approach has identified isopropylmalate synthase 1 (IPMS1), a critical enzyme for Leu biosynthesis, as a modulator of vacuolar morphology. The authors’ in-depth characterization of this mutant revealed that the homeostasis of branched-chain amino acids impacts TOR activity and, thereby, modulates actin cytoskeleton organization. In addition, the authors provide evidence that this mechanism has an immediate influence on plant expansion.

**Decision letter after peer review:**

Thank you for submitting your article "Homeostasis of Branched-Chain Amino Acids Is Critical for the Activity of TOR Signaling in Arabidopsis" for consideration by *eLife*. Your article has been reviewed by three peer reviewers, including Jürgen Kleine-Vehn as the Reviewing Editor, and the evaluation has been overseen by Christian Hardtke as the Senior Editor.

The reviewers have discussed the reviews with one another and the Reviewing Editor has drafted this decision to help you prepare a revised submission.

The reviewers were largely positive about your forward genetic approach and in depth characterization of how the homeostasis of branched-chain amino acids impact on TOR-dependent subcellular processes. However, your manuscript would require substantial revision before being suitable for publication at *eLife*. Mandatory revisions should include:

1) The reviewers did not insist on the genetic interaction data, but encourage you to include it in case the experiments are currently ongoing. On the other hand, they ask you to use the TOR inhibitors in combination with the *ipms1* mutant (e.g. detailed response curves).

2) The reviewers request a thorough discussion of the recent literature – especially highlighting the work of the Ringli lab (e.g. Schaufelberger et al., 2019).

3) They, moreover, urge the author to provide a better explanation/description and quantitative assessment of their data.

4) The effect of BCAA addition on TOR activity (e.g. phosphorylation-specific antibodies) should be investigated, which would further support the conclusions of this paper.

Please, see the more specific comments of the reviewers below, which may guide you to further improve your manuscript.

Reviewer #1:

In a forward genetic screen Cao et al. identified *eva1* mutants, displaying defects in the regulation of actin cytoskeleton, which correlates with several defects on endomembrane organization. The authors reveal that *eva1* mutation disrupts IPMS1 and thereby impact on branched amino acid synthesis, which in turn indirectly modulates the TOR signaling. Moreover, interference with the TOR pathway phenocopies the cellular phenotypes of *eva1*.

The work is of high quality, but largely lacks quantifications and I have several concerns that I would like the authors to address.

Genetic interference with the TOR pathway in *eva1* (by crossing tor-es) would be really nice.

The authors propose that the root organ growth defects in *eva1* is due to altered TOR responses. It would be hence nice to test this hypothesis by gradually reducing TOR activity (concentration range of drugs and inducer for inducible line) in *eva1*. If causal for the root growth defects, these conditions should rescue the root length. A similar approach (using low drug concentrations) could be used to proof the causality for latrunculin B as well as wortmannin. (Please, see Supplementary data in Barberon et al., 2014 where the authors used wortmannin in a similar manner.)

The authors provide a model, assuming that TORC1 and TORC2 is differentially activated by BCAA. This conclusion is based on the fact that *raptor1b* mutants failed to suppress the BCAA-induced bundling of actin. However, the phenotype was not quantitatively addressed. In general, it would be, hence, nice to truly quantify these subcellular phenotypes (lack of quantification is apparent in Figure 1; Figure 1—figure supplements 1 and 2; Figure 4—figure supplement 4; Figure 5; Figure 5—figure supplement 1; Figure 6; Figure 6—figure supplement 1; Figure 7; Figure 7—figure supplement 1). Besides, Raptor 1a is likely redundant and hence the lack of single mutant phenotype is not conclusive.

The authors mention some defects in vacuolar fusion of *eva1* mutants and propose in their model that also the TOR pathway controls vacuolar fusions. These defects are not visible in the provided data and would need additional depiction (and quantifications). In this regard, the authors assume that wortmannin induced vacuolar fusions, which also reverses certain aspects of the *eva1* phenotype. However, it is unclear whether this relates to vacuolar fusion, because this impact could also be due to the (presumably indirect) effect of wortmannin on actin dynamics (see Supplementary data in Scheuring et al., 2016).

Reviewer #2:

This manuscript reports on a new link in plants between the activation of the conserved and important Target of Rapamycin (TOR) kinase and the accumulation of branched-chain amino-acids (BCAA). This link has been well described in yeast and animal cells but was so far not established in plants. These findings are clearly of interest since the mechanisms linking TOR to nutrients are much less characterized in plants then in other eukaryotes.

The authors have used a forward genetic screen for defects in vacuolar morphology in Arabidopsis and have uncovered a mutation in an enzyme (IPMS1) involved in Leu biosynthesis. The authors then showed that disruption in BCAA homeostasis through *ipms1* mutations result in pleiotropic defects in AA metabolism, growth and developments of the mutant lines as well as perturbations in ER and actin networks. Finally, these defects were linked to activation of the TOR signalling pathway by pharmaceutical and genetic means.

The findings in this manuscript are novel, and the claims are most of the time well supported by the solid and convincing experimental results. This paper is very well presented and easy to read.

Nevertheless, I have a few issues which would need to be resolved by the authors.

1) In a very recent paper, Schaufelberger et al., 2019, found that a mutation in IPMS1 suppresses the *lrx1* phenotype (aberrant root hairs) that is also suppressed by TOR. This *ipms1* allele, called *rol17*, shows reduced sensitivity to the TOR inhibitor AZD8055 suggesting higher TOR activity. I believe that this publication should be cited and discussed. For example the effect of *ipms1* mutation on root hair morphology in response to TOR is already described in this published article, although in a different genetic context. By the way, the JEB special issue in which this paper appeared also provides some more recent reviews on the TOR pathway in plants which could be added in the Introduction section.

2) To better support their claims, I think that the authors could explore the effect of BCAA supplied to the plant growth medium on TOR activity.

3) In Figure 7, the authors convincingly show that TOR activity is needed to observe the effects of BCAA on actin organization. I wonder how was the "red line" defined and positioned? Are the fluorescence profiles consistent across the images? Maybe some kind of statistical analysis would be needed here to support the authors' conclusions.

4) From the analysis of a raptor mutant (a specific component of the TORC1 complex) in Figure 7, the authors concluded that the impact of BCAA is not linked to TORC1 but maybe to a putative plant TORC2 complex. While the TORC2 complex has been identified and well described in yeast and animals, the existence of such a complex in plants remains elusive. Moreover, there are two Raptor genes in Arabidopsis and the single raptor1 mutants displays limited phenotypic changes, while the raptor double mutants are more severely affected (see Anderson et al., 2005). Double raptor mutants could be used here but are barely viable. I think that authors should be more cautious in their conclusions on the role of TORC1 in BCAA activation of TOR. Moreover, Son et al. and Ericksen et al. recently showed (Cell Metabolism 2019) that Leu and BCAA effect is TORC1-dependent.

Reviewer #3:

The analysis of *eva1* as a modulator of actin and vacuole development is interesting. The connection to TOR is interesting as well. However, the authors must have overseen the publication by Schaufelberger et al., 2019, demonstrating that mutations in IPMS1 influence TOR. Thus, the novelty of an IMPS1 mutation affecting the TOR network is not new; the type of data are still new since this other publication used a completely different approach and made other analyses. It is important, however, that the authors address this paper and take it into consideration in their Discussion. Several times in the manuscript, it is difficult for a non-specialist reader to understand how the analyses were done (e.g. Figure 7) and the conclusions were drawn.

The following points should be addressed (there are neither page nor line numbers, which makes it difficult to indicate the position in the text!):

1) The Abstract is not ideal; what does glucose have to do with BCAA (considering that at this point, the reader doesn't know about BCAA's role in TOR)

2) The Introduction should contain information on vacuole development and introduce the experimental set-up. What are TVSs? What is the GFP-TIP line? What construct was used? Alternatively, the screen should be introduced at the beginning of the Results section. It seems that the EMS screen was not done with the GFP-TIP expressing line? This is unclear.

3) The Introduction should also introduce the Schaufelberger et al. paper since it has direct implications on the results presented.

4) According to the data shown, *eva1* roots have more cells in the different zones, longer zones, longer cells, and increased meristem activity. How is it possible that the roots are shorter than the wild type?

5) In "loss of function of IPMS1….": homotypic vacuoles and F-actin bundling are affected and influenced by TOR, but there is no evidence that this is directly caused by the altered Leu biosynthesis. In fact, the authors show that TOR is the link between Leu biosynthesis and these observed cellular effects.

6) Explain, how the experiment on the retention of SEC-RFP was done.

7) I don't understand Figure 7. what are the graphs? The explanation in the legend doesn't clarify. Please re-formulate

8) The Discussion remains rather superficial and does not really go into the details of the findings or just don't mention them again. For example, what do we learn from the change in chloroplast/grana structures? How come that a short root has more and longer cells and a more active meristem? TOR activity seems increased, which would produce longer roots (inhibiting TOR by AZD or Torin treatment causes shorter roots). How do the findings fit with the literature on vacuole development.? Importantly, put it in the context of the Schaufelberger et al. (JXB). There, inhibiting TOR mimics the effect of the mutation in *ipms1*, which is different from the findings described here. Also, *imps1* and inhibiting TOR by AZD suppresses a mutation in an LRR-extensin gene, encoding a type of protein that influences vacuole development (Dünser et al., 2019). Thus, there is published evidence that support the findings described in this manuscript, and this should be discussed.

---

## [Author Response]

[…] 1) The reviewers did not insist on the genetic interaction data, but encourage you to include it in case the experiments are currently ongoing. On the other hand, they ask you to use the TOR inhibitors in combination with the ipms1 mutant (e.g. detailed response curves).2) The reviewers request a thorough discussion of the recent literature – especially highlighting the work of the Ringli lab (e.g. Schaufelberger et al., 2019).3) They, moreover, urge the author to provide a better explanation/description and quantitative assessment of their data.

Done. As a result the following data charts have been supplemented: Figure 1—figure supplement 1C, D; Figure 1—figure supplement 2F; Figure 3E; Figure 3—figure supplement 2D; Figure 5—figure supplement 1I; Figure 6G, I, K; Figure 6—figure supplements 2 and 3.

4) The effect of BCAA addition on TOR activity (e.g. phosphorylation-specific antibodies) should be investigated, which would further support the conclusions of this paper.

Done. We investigated the effect of BCAA supplementation on TOR kinase activity by following the phosphorylation status of S6K. The revised panels H and I in Figure 6 provide an additional condition of growing wild type on medium containing 1mM BCAAs, which showed increased phosphorylation status of S6K.

Please, see the more specific comments of the reviewers below, which may guide you to further improve your manuscript.Reviewer #1:[…] The work is of high quality, but largely lacks quantifications and I have several concerns that I would like the authors to address.Genetic interference with the TOR pathway in eva1 (by crossing tor-es) would be really nice.The authors propose that the root organ growth defects in eva1 is due to altered TOR responses. It would be hence nice to test this hypothesis by gradually reducing TOR activity (concentration range of drugs and inducer for inducible line) in eva1. If causal for the root growth defects, these conditions should rescue the root length. A similar approach (using low drug concentrations) could be used to proof the causality for latrunculin B as well as wortmannin. (Please, see Supplementary data in Barberon et al., 2014 where the authors used wortmannin in a similar manner.)

We investigated the effects of increasing concentrations of AZD-8055, wortmannin and latrunculin B on primary root elongation. The supplemented experiments and quantification are presented in Figure 5—figure supplement 2 and 3. Figure 5—figure supplement 2 shows that the effect of TOR inhibitor AZD-8055 on *ipms1* primary root elongation is dose-dependent. Lower concentrations (0.1 and 0.2 μM) of AZD-8055 promote growth; however, as plausible because of TOR inhibition, higher concentrations (0.4, 0.6 and 1.0 μM) of AZD-8055 inhibit *ipms1* primary root elongation. By comparison, even a minimal concentration of AZD-8055 inhibits wild type primary root elongation.

As we explained in the manuscript, wortmannin is a PI3K/TOR dual inhibitor, and therefore it may exert multiple effects on plant growth. Barberon et al., 2014 PNAS, Supplementary Figure 3 presents a minimal but significant inhibition of wild type primary root elongation by 1, 10 and 33 μM wortmannin (n ≥ 8). Consistently, our investigation found that 33 μM wortmannin significantly inhibits primary root elongation of wild type and three *ipms1* lines (Figure 5—figure supplement 3C, n = 50 and 100). However, 1 or 10 μM wortmannin did not significantly inhibit wild type primary root elongation. Interestingly, primary root elongation of *ipms1-4* and *ipms1-5* was significantly increased by 1 μM wortmannin, which indicates pharmaceutical inhibition of TOR activity in *ipms1* promotes plant growth, consistent with the AZD-8055.

Despite primary root elongation of wild type and *ipms1* mutants being inhibited by the actin polymerization inhibitor latrunculin B (Lat B), *ipms1* mutants result more resistant to Lat B (Figure 5—figure supplement 3D-F). The previously presented Figure 4—figure supplement 4 proposed the effect of Lat B on later stages of development of the root system (e.g. formation of lateral roots) and, consistent with Figure 5—figure supplement 3D-F, showed that *ipms1* mutants are more resistant to Lat B.

The authors provide a model, assuming that TORC1 and TORC2 is differentially activated by BCAA. This conclusion is based on the fact that raptor1b mutants failed to suppress the BCAA-induced bundling of actin. However, the phenotype was not quantitatively addressed. In general, it would be, hence, nice to truly quantify these subcellular phenotypes (lack of quantification is apparent in Figure 1; Figure 1—figure supplements 1 and 2; Figure 4—figure supplement 4; Figure 5; Figure 5—figure supplement 1; Figure 6; Figure 6—figure supplement 1; Figure 7; Figure 7—figure supplement 1). Besides, Raptor 1a is likely redundant and hence the lack of single mutant phenotype is not conclusive.

In this manuscript, we do not show evidence or propose that potential plant TORC1 and TORC2 are differentially activated. The study of potential differential activation is essentially constrained because TORC2 has not been identified. So far only a single TOR and two TOR interactors, Raptor and LST8, have been identified in plants. In yeast and mammalian models, the Raptor homolog is the defining component of TORC1 and the LST8 homolog exists in both TORC1 and TORC2. Besides, sensors and signaling components of the amino acid sensing pathway upstream of TOR are also unknown in plants.

We used a well-characterized T-DNA insertion mutant of *AtRaptor1B* (also known as *Raptor2*) because of several considerations. Deprost et al., 2005 found disruption of *AtRaptor1A (Raptor1*) has no visible effects on embryo and plant development; Anderson et al., 2005 found *raptor1a* mutant shows almost no changes in their morphology and development; Pu et al., 2017 found *raptor1a* mutant shows minimal defects in autophagy. Therefore, later studies only used *raptor1b* mutant to characterize the functions of TORC1(Salem et al., 2017; Salem et al., 2018; Wang et al., 2018). Besides, the *raptor1ab* double mutant is possibly not informative, because its growth is severely affected and barely viable after embryogenesis (Anderson et al., 2005). Moreover, it is important to consider the multiple layers of reciprocal regulation between TORC1 and TORC2 in yeast and mammalian cells (covered by reviews of Saxton and Sabatini, 2017, and Xie et al., 2018). Therefore, it may not be possible to find a situation, such as the *raptor1ab* double mutant, where TORC1 is completely disabled, while TORC2 is still not affected.

To address your point on quantification, we revised the following figures. Figure 1A and Figure 1—figure supplement 1: To provide quantitative data to describe the vacuolar phenotypes of *eva1/ipms1*, we measured the number of unfused vacuoles and the length of trans-vacuolar strands (TVSs) in cells expressing the tonoplast marker GFP-δTIP. We clarified that we defined unfused vacuoles as spherical structures that are isolated from the large central vacuole and with diameter > 5 μm. Notably, the TVSs were rarely, if ever, observed in 10-day old cotyledon cells of wild type, but very abundant in the mutant. We underlined this marked differences in the text and legend relevant to Figure 1. Because of the obvious phenotype, we did not quantify TVSs in the later figures.

In Figure 1—figure supplement 1, we used γTIP-YFP to show the *eva1* vacuolar phenotypes are not an artefact due to the choice of GFP-δTIP marker. However, compared to GFP-δTIP, γTIP-YFP labeled numerous additional vacuolar structures. The nature of these vacuolar structures is still not well understood (Gattolin et al., 2010), and the unfused vacuoles are not clearly shown by γTIP-YFP. Therefore, we did not see the necessity to quantify the TVSs and but we provided magnified images showing the enhanced TVSs in *eva1* background for comparison to wild type.

Figure 1—figure supplement 2: We quantified the number of unfused vacuoles in each cell.

Figure 4—figure supplement 4 presents photographs showing primary root elongation on Lat B medium, which are already accompanied with quantification of primary root elongation.

Figure 5, Figure 5—figure supplement 1: We measured the number of unfused vacuoles in each cell, in wild type and in mutant, before and after inhibitor treatment. A chart was added to the figure supplement.

Figure 6 and Figure 6—figure supplement 1: We quantified the number of unfused vacuoles, and a chart was added as Figure 6G. We also quantified S6K-p/S6K ratio following an example of Dong et al., 2017, and a chart was added to the figure.

Figure 6I. We quantified EdU intensity using relative unit of fluorescence intensity, and the chart was added as Figure 6K.

Figure 7, Figure 7—figure supplement 1: We revised the figure legends to clarify the experimental setup and the interpretation that show strikingly induced actin bundling.

The authors mention some defects in vacuolar fusion of eva1 mutants and propose in their model that also the TOR pathway controls vacuolar fusions. These defects are not visible in the provided data and would need additional depiction (and quantifications). In this regard, the authors assume that wortmannin induced vacuolar fusions, which also reverses certain aspects of the eva1 phenotype. However, it is unclear whether this relates to vacuolar fusion, because this impact could also be due to the (presumably indirect) effect of wortmannin on actin dynamics (see Supplementary data in Scheuring et al., 2016).

Our data support that unfused vacuole phenotype is due to TOR activation. First, the unfused vacuole phenotype is observed in *ipms1* mutants (Figure 1A; Figure 1—figure supplement 2D) and wild type growing with BCAA supplementation (Figure 7—figure supplement 3), which we showed are associated with up-regulated TOR signaling. Furthermore, the unfused vacuole phenotype is recovered by TOR inhibitor AZD-8055 and Torin2 (Figure 6A-F; Figure 6—figure supplement 1; Figure 7—figure supplement 3). These results have been discussed in light of a large number of recent publications in yeast and mammalian cells, which indicate that the dynamic fusion, fission and size of vacuoles and lysosomes are controlled by TOR signaling through yet unknown mechanisms. We included discussion of two representative publications (Yu et al., 2010; Michaillat et al., 2012) showing that depending on the nutrient status TORC1 regulates the size and number of yeast vacuoles and mammalian lysosomes (subsection “Plant endomembrane homeostasis is correlated with TOR activity”).

Wortmannin inhibits PI3K and PI4K activity and phospholipid synthesis in plant cells (Matsuoka et al., 1995 JCB), and therefore it directly effects the membrane fusion machinery (Wickner, 2010 Annu Rev Cell Dev Biol). Wortmannin-induced vacuole fusion has been demonstrated by several studies in plant cells (Wang et al., 2009; Cui et al., 2014; Alvarez et al., 2016 JXB; Zheng et al., 2014). Importantly, Wm and LY294002 are effective inhibitors of mammalian TOR (Brunn et al., 1996) and are considered as PI3K/TOR dual inhibitors (Benjamin et al., 2011). Our work supports that caution should be taken when interpreting wortmannin’s effects on vacuole and actin in the study of membrane traffic in general. Indeed, our work supports that the role of TOR as master regulator of actin organization is conserved.

Reviewer #2:[…] I have a few issues which would need to be resolved by the authors.1) In a very recent paper, Schaufelberger et al., 2019) found that a mutation in IPMS1 suppresses the lrx1 phenotype (aberrant root hairs) that is also suppressed by TOR. This ipms1 allele, called rol17, shows reduced sensitivity to the TOR inhibitor AZD8055 suggesting higher TOR activity. I believe that this publication should be cited and discussed. For example the effect of ipms1 mutation on root hair morphology in response to TOR is already described in this published article, although in a different genetic context. By the way, the JEB special issue in which this paper appeared also provides some more recent reviews on the TOR pathway in plants which could be added in the Introduction section.

We highlighted the work of Schaufelberger et al., 2019, in the Introduction, where we first introduced the concepts of BCAAs, IPMS and TOR (Introduction, third paragraph). Additionally, we thoroughly discussed the results in Schaufelberger et al., 2019, in the first section of Discussion (and subsection “Plant endomembrane homeostasis is correlated with TOR activity”).

2) To better support their claims, I think that the authors could explore the effect of BCAA supplied to the plant growth medium on TOR activity.

We investigated the effects of BCAA supplementation on TOR kinase activity by detecting the phosphorylation status of S6K. The revised Figure 6H and I provide an additional condition of growth of wild type on medium containing 1mM BCAAs, which showed increased phosphorylation status of S6K.

3) In Figure 7, the authors convincingly show that TOR activity is needed to observe the effects of BCAA on actin organization. I wonder how was the "red line" defined and positioned? Are the fluorescence profiles consistent across the images? Maybe some kind of statistical analysis would be needed here to support the authors' conclusions.

We are grateful for the appreciation of an evident connection between the actin reorganization phenotype and TOR activation, which highlights a role of TOR as a conserved master regulator of actin organization. To our knowledge, such a drastic reorganization has been reported in mammalian cultured cells upon TOR stimulation, but has not been reported in intact organisms. In order to clarify the experimental setup and the interpretation, we revised the description in figure legends. Higher fluorescence intensity of the actin marker suggests more bundling of actin filaments. Using ImageJ, we drew a 50 μm arrowed line to detect the fluorescence intensity of the pixels covered by such a line. In each image, the arrowed line is positioned where we detected the highest fluorescence intensity in non-saturating imaging conditions. A chart beneath the image presents plotted fluorescence intensity along the red arrowed line. Without actin bundling, fine actin filaments have fluorescence intensity about 1000 (relative units). In contrast, induced actin bundling show fluorescence intensity peaks of 3000 – 4000 (relative units). The images were acquired with the same confocal image settings of laser, detectors and pinhole aperture.

4) From the analysis of a raptor mutant (a specific component of the TORC1 complex) in Figure 7, the authors concluded that the impact of BCAA is not linked to TORC1 but maybe to a putative plant TORC2 complex. While the TORC2 complex has been identified and well described in yeast and animals, the existence of such a complex in plants remains elusive. Moreover, there are two Raptor genes in Arabidopsis and the single raptor1 mutants displays limited phenotypic changes, while the raptor double mutants are more severely affected (see Anderson et al., 2005). Double raptor mutants could be used here but are barely viable. I think that authors should be more cautious in their conclusions on the role of TORC1 in BCAA activation of TOR. Moreover, Son et al. and Ericksen et al. recently showed (Cell Metabolism 2019) that Leu and BCAA effect is TORC1-dependent.

We used a well-characterized T-DNA insertion mutant of *AtRaptor1B* (also known as

*Raptor2*) because of several considerations. Deprost et al., 2005 found disruption of *AtRaptor1A (Raptor1*) has no visible effects on embryo and plant development, and Anderson et al., 2005 found *AtRaptor1A* T-DNA lines show almost no changes in their morphology and development. However, we agree that the *raptor1ab* double mutant would not be informative because its growth is severely affected and it is barely viable after embryogenesis (Anderson et al., 2005). Moreover, it is important to consider the multiple layers of reciprocal regulation between

TORC1 and TORC2 in yeast and mammalian cells (covered by reviews of Saxton and Sabatini, 2017, and Xie et al., 2018). Therefore, it may not be possible to find a situation, such as the *raptor1ab* double mutant, where TORC1 is completely disabled, while TORC2 is still not affected.

We appreciate your suggestions about the two recent papers of Cell Metabolism. Son et al., found that a metabolite of Leu promotes Raptor acetylation and mTORC1 activation. Ericksen et al., found that loss of BCAA catabolism enhances mTORC1 activity during tumor development. These two papers provide important evidence supporting that amino acid sensing is upstream of mTORC1. However, considering the complexity of amino acid sensing mechanisms (e.g., various sensors for specific amino acids and signaling components, mostly reviewed by Kim and Guan, 2019), BCAAs may also signal to mTORC2 in a mTORC1-independent manner. In two seminal studies of mTORC2, Sarbassov et al., 2004, found Rictor and mTOR, but not Raptor, are required for PKC phosphorylation and actin organization; Jacinto et al., 2004, found nutrient repletion after starvation stimulates actin organization. Nevertheless, much less is known about the activation mechanism of mTORC2 than that of mTORC1. Therefore, we discussed this issue in the manuscript and added a question mark for TORC2 activation by BCAAs in the proposed model (Figure 8).

Reviewer #3:The analysis of eva1 as a modulator of actin and vacuole development is interesting. The connection to TOR is interesting as well. However, the authors must have overseen the publication by Schaufelberger et al., 2019, demonstrating that mutations in IPMS1 influence TOR. Thus, the novelty of an IMPS1 mutation affecting the TOR network is not new; the type of data are still new since this other publication used a completely different approach and made other analyses. It is important, however, that the authors address this paper and take it into consideration in their Discussion. Several times in the manuscript, it is difficult for a non-specialist reader to understand how the analyses were done (e.g. Figure 7) and the conclusions were drawn.The following points should be addressed (there are neither page nor line numbers, which makes it difficult to indicate the position in the text!):

Thank you for your suggestions on the format. We added line numbers and page numbers for the convenience of your reviewing process. The other suggestions also have been carefully addressed as the followings.

1) The Abstract is not ideal; what does glucose have to do with BCAA (considering that at this point, the reader doesn't know about BCAA's role in TOR)

To make a clear introduction, we revised “a functional connection of TOR with glucose availability has been demonstrated” to “a functional connection of TOR activation with glucose availability has been demonstrated” (line 12).

2) The Introduction should contain information on vacuole development and introduce the experimental set-up. What are TVSs? What is the GFP-TIP line? What construct was used? Alternatively, the screen should be introduced at the beginning of the Results section. It seems that the EMS screen was not done with the GFP-TIP expressing line? This is unclear.

We appreciate your suggestion on the necessity of introducing plant vacuoles at the beginning of the manuscript. A paragraph of the revised Introduction (from line 78) is dedicated to introduce plant vacuoles and ER. The forward genetic screen used an EMS-mutagenized GFPδTIP line to identify mutants defective in vacuole morphogenesis. To provide more information of GFP-δTIP, we introduced the full name “tonoplast intrinsic protein”. Cutler et al., 2000, generated the GFP-δTIP construct and established it is a vacuolar marker. The original screen of GFP-δTIP labeled vacuolar mutants was carried out by Avila et al., 2003.

3) The Introduction should also introduce the Schaufelberger et al. paper since it has direct implications on the results presented.

We introduced the work of Schaufelberger et al., 2019, in the Introduction, where we first discuss the concepts of BCAAs, IPMS and TOR (line 50). Additionally, we thoroughly discussed the results in Schaufelberger et al., 2019, in the first section of Discussion (and subsection “Plant endomembrane homeostasis is correlated with TOR activity”).

4) According to the data shown, eva1 roots have more cells in the different zones, longer zones, longer cells, and increased meristem activity. How is it possible that the roots are shorter than the wild type?

For *ipms1* mutants, the aberrant development of root tip (e.g., higher activity of root tip meristem and strikingly inhibited formation of root hairs) may contribute to, but certainly does not conflict with, shorter primary root. “Primary root” is the centric and main root, while root tip with three zones is only a small fraction of the primary root. We used the term “primary root length” in text and figures to avoid confusion. Additionally, we delineated the complexity of *ipms1* root tip phenotypes. Despite *ipms1* root tips having longer zones and more cells in each zone compared to wild type, *ipms1* cell length is shorter than wild type, which is shown by a new chart Figure 3— figure supplement 2D.

5) In "loss of function of IPMS1….": homotypic vacuoles and F-actin bundling are affected and influenced by TOR, but there is no evidence that this is directly caused by the altered Leu biosynthesis. In fact, the authors show that TOR is the link between Leu biosynthesis and these observed cellular effects.

Our results and conclusions are not centered on altered Leu biosynthesis. Using BCAA profiling of *ipms1* mutants, other BCAA biosynthetic mutants and exogenous feeding of BCAAs, we revealed a generalized causal relationship between BCAA over-accumulation, up-regulated TOR activity, and reorganization of subcellular structures.

6) Explain, how the experiment on the retention of SEC-RFP was done.

As we introduced in the subsection “The organization of ER network and actin cytoskeleton is altered in *eva1*”, SEC-RFP is a bulk flow marker and its secretion to apoplast indicates proper activity of bulk-flow secretion, which is an important function of secretory pathway and endomembrane system. On the other hand, retention of SEC-RFP inside cells indicates defects in secretion, such as protein sorting in the trans-Golgi network in citation 51 and 52 (Renna et al., 2013; Renna et al., 2018). In both wild type and *eva1*, SEC-RFP is secreted to apoplast and not retained inside cells (Figure 4—figure supplement 3A), posing that although in *eva1* the organization of ER and Golgi is altered, but bulk-flow traffic is not affected. This has been verified also in other mutants, such as *gom8/rhd3-7* (Stefano et al., 2012) and *g92/sec24b* (Faso et al., 2009).

7) I don't understand Figure 7. what are the graphs? The explanation in the legend doesn't clarify. Please re-formulate

We clarified the experimental setup and the interpretation of the results. Higher fluorescence intensity of the actin marker indicates more bundling of actin filaments. Using ImageJ, we drew a 50 μm arrowed line to detect the pixel fluorescence intensity beneath such a line. In each image, the arrowed line is positioned where we detected the highest fluorescence intensity using non-saturating imaging settings. A chart beneath the image presents plotted fluorescence intensity along the red arrowed line. Without actin bundling, fine actin filaments have fluorescence intensity about 1000 (relative unit). In contrast, induced actin bundling show fluorescence intensity peaks of 3000 – 4000 (relative unit). The images were acquired with the same confocal image settings of laser, detectors and pinhole aperture.

8) The Discussion remains rather superficial and does not really go into the details of the findings or just don't mention them again. For example, what do we learn from the change in chloroplast/grana structures? How come that a short root has more and longer cells and a more active meristem? TOR activity seems increased, which would produce longer roots (inhibiting TOR by AZD or Torin treatment causes shorter roots). How do the findings fit with the literature on vacuole development.? Importantly, put it in the context of the Schaufelberger et al., 2019. There, inhibiting TOR mimics the effect of the mutation in ipms1, which is different from the findings described here. Also, imps1 and inhibiting TOR by AZD suppresses a mutation in an LRR-extensin gene, encoding a type of protein that influences vacuole development (Dünser et al., 2019). Thus, there is published evidence that support the findings described in this manuscript, and this should be discussed.

We appreciate your encouragement to discuss all *ipms1* phenotypes. To our knowledge the subcellular characterization proposed in our work is novel and establishes *ipms1* as a model to study up-regulation of TOR signaling in planta at a cell and organ level. Therefore, we provided further discussion, as recommended. Here, we briefly summarize the new points as the follows. One of the most notable *ipms1* plant growth phenotypes is the increase in leaf thickness and alteration of chloroplast ultrastructure, which is similar to a knockdown mutant of isopropylmalate isomerase (IPMI), the enzyme following IPMS for Leu biosynthesis (Imhof et al., 2014). These results support an association of BCAA accumulation with defective chloroplast development. Loss of stroma thylakoid was reported in a mutant defective in photosystem I assembly (Liu et al., 2012 Plant Cell), but the underlying mechanism is unclear.

Furthermore, our data indicate that TOR activity in *ipms1* mutants is higher than wild type. In *ipms1* mutants, an inconsistency between nutrient status and the activity of TOR metabolic signaling is detrimental to plant growth and development. A supplemented measurement of primary root length on medium with increasing concentrations of AZD-8055 provides direct evidence, because low concentrations of AZD-8055 promote *ipms1* primary root elongation (Figure 5—figure supplement 2). We also found that abrupt activation of TOR affects root development possibly through actin organization, because *ipms1* mutants show severely delayed formation root hairs (Figure 3—figure supplement 2) and defective development of overall root system, despite of a higher activity of root apical meristem.

We also discussed our findings about vacuole morphogenesis and root development in light of the reports by Schaufelberger et al., 2019, and Dünser et al., 2019, in the context of BCAA activating TOR and TOR regulating vacuole morphology (Discussion, first paragraph and subsection “Plant endomembrane homeostasis is correlated with TOR activity”).